# MODIFICATION-CONSIDERING VALUE LEARNING FOR REWARD HACKING MITIGATION IN RL

## ABSTRACT

Reinforcement learning (RL) agents can exploit unintended strategies to achieve high rewards without fulfilling the desired objectives, a phenomenon known as reward hacking. In this work, we examine reward hacking through the lens of General Utility RL, which generalizes RL by considering utility functions over entire trajectories rather than state-based rewards. From this perspective, many instances of reward hacking can be seen as inconsistencies between current and updated utility functions, where the behavior optimized for an updated utility function is poorly evaluated by the original one. Our main contribution is Modification-Considering Value Learning (MC-VL), a novel algorithm designed to address this inconsistency during learning. Starting with a coarse yet value-aligned initial utility function, the MC-VL agent iteratively refines this function based on past observations while considering the potential consequences of updates. This approach enables the agent to anticipate and reject modifications that may lead to undesired behavior. To validate our approach, we implement MC-VL agents based on the Double Deep Q-Network (DDQN) and Twin Delayed Deep Deterministic Policy Gradients (TD3), demonstrating their effectiveness in preventing reward hacking in diverse environments, including those from AI Safety Gridworlds and the MuJoCo gym.

## 1 INTRODUCTION

From mastering video games (Mnih et al., 2015) to optimizing robotic control (Levine et al., 2016), reinforcement learning (RL) agents have solved a wide range of tasks by learning to maximize cumulative rewards. However, this reward-maximization paradigm has a significant flaw: agents may exploit poorly defined or incomplete reward functions, leading to a behavior known as *reward hacking* (Skalse et al., 2022), where the agent maximizes the reward signal but fails to meet the designer's true objectives.

For instance, an RL agent tasked with stacking blocks instead flipped them, exploiting a reward based on the height of the bottom face of a block (Popov et al., 2017). Similarly, a robot arm manipulated objects in arbitrary ways that exploited a classifier-based reward system, tricking it into labeling incorrect actions as successful due to insufficient negative examples (Singh et al., 2019). Ibarz et al. (2018) describe reward model exploitation in Atari games, where agents exploit flaws in reward functions learned from human preferences and demonstrations. These incidents underscore that while RL agents may maximize rewards, their learned behaviors often diverge from the goals intended by the reward designers.

As RL systems scale to more complex, safety-critical applications like autonomous driving (Kiran et al., 2021) and medical diagnostics (Ghesu et al., 2017), ensuring reliable and safe agent behavior becomes increasingly important. Pan et al. (2022) showed that reward hacking becomes more common as models grow in complexity. Moreover, Denison et al. (2024) demonstrated that agents based on large language models, trained with outcome-based rewards, can generalize to changing the code of their own reward functions. Reward hacking also becomes more prominent with increased reasoning capabilities. For example, during testing of the o1-preview (pre-mitigation) language model on a Capture the Flag (CTF) challenge, the model encountered a bug that prevented the target container from starting. Rather than solving the challenge as intended, the model used nmap to scan the network, discovered a misconfigured Docker daemon API, and exploited it to start the container and read the flag via the Docker API, bypassing the original task altogether (OpenAI, 2024).

In this paper, we frame reward hacking within the General Utility RL (GU-RL) formalism (Zahavy et al., 2021; Geist et al., 2022). We describe an agent that optimizes a learned utility function, which assigns value to trajectories based on past observed rewards. Many instances of reward hacking, such as manipulating the reward provision process (Everitt et al., 2021) and tampering with the sensors (Ring & Orseau, 2011), can be viewed as inconsistent updates to the utility function. We define an update as inconsistent when the trajectories produced by a policy optimized for the updated utility function would be evaluated poorly by the prior utility function. To address this issue, we introduce *Modification-Considering Value Learning* (MC-VL). In MC-VL, the agent updates its utility function based on the observed rewards, similar to value-based RL, but it also predicts the long-term consequences of potential updates and can reject them. In our formulation, avoiding inconsistent utility updates is an optimal behavior.

For example, consider a robot trained to grasp objects using human feedback (Christiano et al., 2017). A standard RL agent, if rewarded for positioning its manipulator between the object and the camera in the middle of the training, can exploit this reward by learning to repeat that behavior (OpenAI, 2017). In contrast, an MC-VL agent would first forecast the consequences of updating its utility function based on this new reward. Drawing from prior experiences where positive rewards were given only for positioning the manipulator near the object, the MC-VL agent might predict low utility for positioning the manipulator in front of the camera. As a result, the agent would reject the update, staying focused on the intended grasping task.

Several prior works have discussed the theoretical possibility of mitigating reward or sensor tampering using *current utility optimization*, where an agent evaluates potential changes to its utility function using its current utility function (Orseau & Ring, 2011; Hibbard, 2012; Everitt et al., 2016; 2021). Dewey (2011) suggested learning the utility function from past observations. However, to the best of our knowledge, no prior work has formalized this within the GU-RL framework, applied this idea to standard RL environments, or implemented such an agent. In this work, we provide an algorithm to learn the utility function, estimate future policies, and compare them using the current utility function. Additionally, we introduce a learning setup where the initial utility function is learned in a *Safe* sandbox version of the environment before transitioning to the *Full* version. Our experiments, conducted across various environments, including benchmarks adapted from the AI Safety Gridworlds (Leike et al., 2017), are, to the best of our knowledge, the first to demonstrate the ability to prevent reward hacking in these environments. Furthermore, our results provide insights into the key parameters influencing MC-VL performance, laying the groundwork for further research on preventing reward hacking in RL.

## 2 RELATED WORK

The problem of agents learning unintended behaviors by exploiting misspecified training signals has been extensively discussed in the literature as *reward hacking* (Skalse et al., 2022), *reward gaming* (Leike et al., 2018), or *specification gaming* (Krakovna et al., 2020). Krakovna et al. (2020) provide a comprehensive overview of these behaviors across RL and other domains. The theoretical foundations for understanding reward hacking are explored by Skalse et al. (2022), who argue that preventing reward hacking requires either limiting the agent's policy space or carefully controlling the optimization process.

Laidlaw et al. (2023) propose addressing reward hacking by regularizing the divergence between the occupancy measures of the learned policy and a known safe policy. Unlike their approach, which may overly restrict the agent's ability to learn effective policies, our method does not require the final policy to remain close to any predefined safe policy. Eisenstein et al. (2024) investigate whether ensembles of reward models trained from human feedback can mitigate reward hacking, showing that while ensembles reduce the problem, they do not completely eliminate it. To avoid additional computational overhead, we do not use ensembles in this work, but they could complement our method by improving the robustness of the learned utility function.

A specific form of reward hacking, where an agent manipulates the mechanism by which it receives rewards, is known as *wireheading* (Amodei et al., 2016; Taylor et al., 2016; Everitt & Hutter, 2016; Majha et al., 2019) or *reward tampering* (Kumar et al., 2020; Everitt et al., 2021). Related phenomena, where an agent manipulates its sensory inputs to deceive the reward system, are discussed as *delusion-boxing* (Ring & Orseau, 2011), *measurement tampering* (Roger et al., 2023), and *reward-input*

*tampering* (Everitt et al., 2021). Several studies have hypothesized that current utility optimization could mitigate reward or sensor tampering (Yudkowsky, 2011; Yampolskiy, 2014; Hibbard, 2012). One of the earliest discussions of this issue is in by Schmidhuber (2003), who developed the concept of *Gödel-machine* agents, capable of modifying their own source code, including the utility function. They suggested that such modifications should only occur if the new values are provably better according to the old ones. However, none of these works addressed learning the utility function or described the optimization process in full detail.

Dewey (2011) introduced the concept of *Value-Learning Agents*, which learn and optimize a utility function based on past observations as a potential solution to reward tampering. Everitt & Hutter (2016) considered a setting where the agent learns a posterior given a prior over manually specified utility functions, proposing an agent that is not incentivized to tamper with its reward signal by selecting actions that do not alter its beliefs about the posterior. More recently, Everitt et al. (2021) formalized conditions under which an agent optimizing its current reward function would lack the incentive to tamper with the reward signal. Our work suggests an implementation of value learning in standard RL environments, where the utility function is learned from the past rewards. Additionally, our method is applicable to other instances of reward hacking beyond reward tampering. Moreover, it aims to prevent reward hacking, rather than simply removing the incentive for it.

## 3    BACKGROUND

We consider the usual Reinforcement Learning (RL) setup, where an agent learns to make decisions by interacting with an environment and receiving feedback in the form of rewards (Sutton & Barto, 2018). This interaction is modeled as a Markov Decision Process (MDP) (Puterman, 2014) defined by the tuple $(S, A, P, R, \rho, \gamma)$, where $S$ is the set of states, $A$ is the set of actions, $P : S \times A \times S \to \mathbb{R}$ is the transition kernel, $R : S \times A \to \mathbb{R}$ is the reward function, $\rho$ is the initial state distribution, and $\gamma$ is the discount factor. The objective in a standard RL is to learn a policy $\pi : S \to A$ that maximizes the expected return, defined as the cumulative discounted reward $\mathbb{E}_\pi \left[ \sum_{t=0}^\infty \gamma^t R(s_t, a_t) \right]$. The expected return from taking action $a$ in state $s$ and subsequently following policy $\pi$ is called state-action value function and denoted as $Q^\pi(s, a)$.

**Deep Q-Networks (DQN) and Double DQN (DDQN)**    DQN (Mnih et al., 2013) and DDQN (van Hasselt et al., 2016) are RL algorithms that approximate the state-action value function $Q(s, a; \theta)$ using neural networks, where $\theta$ are the network parameters. Both algorithms store past experiences in a replay buffer and update network parameters by minimizing a loss $\mathcal{L}(\theta)$ on the temporal-difference error based on the Bellman equation:

$$\mathcal{L}(\theta) = ||Q(s_t, a_t; \theta) - sg[r_t + \gamma Q(s_{t+1}, \arg\max_a Q(s_{t+1}, a; \hat{\theta}); \theta^-)]||, \tag{1}$$

where $sg$ denotes stop gradient, $(s_t, a_t, r_t, s_{t+1})$ represents a transition sampled from the buffer, and $\theta^-$ refers to parameters of a target network, which stabilizes learning by being a slower updating version of the current Q-network. DQN uses $\hat{\theta}$ equal to $\theta^-$, while DDQN proposed to use $\theta$ instead to reduce the overestimation bias. The policy $\pi(s)$ is obtained by $\arg\max_a Q(s, a; \theta)$.

**General-Utility RL (GU-RL)**    In this work, we focus on an agent that optimizes its current utility function. This problem is naturally framed within the General-Utility Reinforcement Learning (GU-RL) (Geist et al., 2022; Zhang et al., 2020; Zahavy et al., 2021), which generalizes standard RL to maximization of utility function $F$. Unlike traditional RL, where rewards are assigned to individual transitions, $F$ intuitively assigns value to entire trajectories. GU-RL offers a more general framework that encompasses tasks like risk-sensitive RL (Mihatsch & Neuneier, 2002), apprenticeship learning (Abbeel & Ng, 2004), and pure exploration (Hazan et al., 2019).

Formally, the utility function $F$ maps an occupancy measure to a real value. An occupancy measure describes the distribution over state-action pairs encountered under a given policy. For a given policy $\pi$ and an initial state distribution $\rho$, the occupancy measure $\lambda_\rho^\pi$ is defined as

$$\lambda_\rho^\pi(s, a) \overset{\text{def}}{=} \sum_{t=0}^{+\infty} \gamma^t \mathbb{P}_{\rho, \pi}(s_t = s, a_t = a),$$

where $\mathbb{P}_{\rho,\pi}(s_t = s, a_t = a)$ is the probability of observing the state-action pair $(s, a)$ at time step $t$ under policy $\pi$ starting from $\rho$. The utility function $F(\lambda_\rho^\pi)$ assigns a scalar value to the occupancy measure induced by the policy $\pi$. The agent's objective is to find a policy $\pi$ that maximizes $F(\lambda_\rho^\pi)$.

A trajectory $\tau = (s_0, a_0, \ldots, s_h, a_h)$ induces the occupancy measure $\lambda(\tau)$, defined as

$$\lambda(\tau) \overset{\mathrm{def}}{=} \sum_{t=0}^{h} \gamma^t \delta_{s_t, a_t},$$

where $\delta_{s,a}$ is an indicator function that is 1 only for the state-action pair $(s, a)$ (Barakat et al., 2023).

Standard RL is a special case of GU-RL, where the utility function $F_{RL}$ is linear with respect to the occupancy measure, and maximizing it corresponds to maximizing the expected cumulative return:

$$F_{RL}(\lambda_\rho^\pi) = \langle R, \lambda_\rho^\pi \rangle = \mathbb{E}_\pi \left[ \sum_{t=0}^{\infty} \gamma^t R(s_t, a_t) \right].$$

## 4 METHOD

We aim to address reward hacking in RL by introducing *Modification-Considering Value Learning* (MC-VL). The MC-VL agent continuously updates its utility function based on observed rewards while avoiding inconsistent utility modifications that could lead to suboptimal behavior under the current utility function. This is achieved by comparing policies induced by the current and updated utility functions. To compare the policies, we compare the trajectories they produce.

**Trajectory Value Function**   We introduce *trajectory value functions* to compute the values of the trajectories produced by the policies. A trajectory value function $U^\pi(\tau)$ evaluates the utility of an occupancy measure induced by starting with a trajectory $\tau = (s_0, a_0, \ldots, s_h, a_h)$ and following a policy $\pi$ after the end of this trajectory:

$$U^\pi(\tau) \overset{\mathrm{def}}{=} F\left( \lambda(\tau) + \gamma^h \lambda_{S_{h+1}}^\pi \right),$$

where $S_{h+1}$ is the distribution of the states following the $\tau$, and $\lambda_{S_{h+1}}^\pi$ represents the occupancy measure induced by following $\pi$ from $S_{h+1}$. In the standard RL setting, this simplifies to the following:

$$U_{RL}^\pi(\tau) = \langle R, \lambda(\tau) + \gamma^h \lambda_{S_{h+1}}^\pi \rangle = \sum_{t=0}^{h-1} \gamma^t R(s_t, a_t) + \gamma^h Q^\pi(s_h, a_h).$$

Every trajectory value function has a corresponding utility function $F(\lambda_\rho^\pi) = \mathbb{E}_{\tau \in \mathcal{T}_\rho^\pi} U^\pi(\tau)$, where $\mathcal{T}_\rho^\pi$ denotes a distribution of trajectories started from state distribution $\rho$ and continued by following a policy $\pi$. Thus, it is also referred to as *utility* for brevity.

**General Utility Generalized Policy Iteration (GU-GPI)**   To formalize a learning process using the trajectory value functions, we extend Generalized Policy Iteration (GPI) (Sutton & Barto, 2018) to the general utility setting, resulting in *General Utility Generalized Policy Iteration* (GU-GPI). In GU-GPI, the algorithm alternates between refining the value estimates of trajectories and improving the policy toward maximizing this value. Specifically, at each time step $t$:

$$U_t \rightsquigarrow U^{\pi_{t-1}}, \quad \pi_t \rightsquigarrow \arg\max_\pi \mathbb{E}_{\tau \sim \mathcal{T}_\rho^\pi} U^\pi(\tau).$$

**Value Learning (VL)**   The *value-learning* agent optimizes a utility $U_{VL}$, which is learned from observed transitions (Dewey, 2011). Algorithm 1 provides the GU-GPI for a value learning agent. In our framework, the agent begins with an initial utility $U_{VL_0}$, and updates it towards the RL-based utility $U_{RL}$ after each environment step, using trajectories $\mathcal{T}(D)$ formed from the set of previously observed transitions $D$:

$$\mathcal{T}(D) = \{(s_0, a_0, \ldots, s_h, a_h) \, \forall t \in \{0, \ldots, h-1\} \, \exists (s, a, s', r) \in D \text{ s.t. } (s_t, a_t, s_{t+1}) = (s, a, s')\}.$$

**Algorithm 1** Value-Learning (VL)

**Input**: Replay buffer $D$, policy $\pi_0$, and initial utility $U_{VL_0}$

1: **for** time step $t = 0$, while not converged **do**
2:     $U_{t+1} \rightsquigarrow U_{VL_t}^{\pi_t}$                      ▷ Update $U$
                                                   ▷ Improve $\pi$
3:     $\pi_{t+1} \rightsquigarrow \arg\max_\pi \mathbb{E}_{\tau \sim \mathcal{T}_\rho^\pi}[U_{t+1}(\tau)]$
4:     $a_t \leftarrow \pi_t(s_t)$                      ▷ Select action
5:     Update utility:

   $D \leftarrow D \cup \{T_{t-1}\}$

   $U_{VL_{t+1}}^{\pi_{t+1}}(\tau) \rightsquigarrow U_{RL}^{\pi_{t+1}}(\tau) \mid \tau \in \mathcal{T}(D)$

6:     $s_{t+1}, r_t \leftarrow act(a_t)$          ▷ Perform action
7:     $T_t \leftarrow (s_t, a_t, s_{t+1}, r_t)$
8: **end for**

**Algorithm 2** Modification-Considering VL

**Input**: Replay buffer $D$, policy $\pi_0$, and initial utility $U_{VL_0}$

1: **for** time step $t = 0$, while not converged **do**
2:     $U_{t+1} \rightsquigarrow U_{VL_t}^{\pi_t}$                      ▷ Update $U$
                                                   ▷ Improve $\pi$
3:     $\pi_{t+1} \rightsquigarrow \arg\max_\pi \mathbb{E}_{\tau \sim \mathcal{T}_\rho^\pi}[U_{t+1}(\tau)]$
4:     $(a_t, modify) \leftarrow \pi_t(T_{t-1})$
5:     **if** $modify$ **then**

   $D \leftarrow D \cup \{T_{t-1}\}$

   $U_{VL_{t+1}}^{\pi_{t+1}}(\tau) \rightsquigarrow U_{RL}^{\pi_{t+1}}(\tau) \mid \tau \in \mathcal{T}(D)$

6:     **end if**
7:     $s_{t+1}, r_t \leftarrow act(a_t)$          ▷ Perform action
8:     $T_t \leftarrow (s_t, a_t, s_{t+1}, r_t)$
9: **end for**

Q-learning algorithms such as DQN or DDQN can be seen as special cases of the value-learning agent, where $U_{t+1}$ is updated to be an exact copy of $U_{VL_t}^{\pi_t}$, and $U_{VL_t}^{\pi_t}$ only learns the state-action value of the first state and action in a trajectory: $U_{VL_t}^{\pi_t}(s_0, a_0, \ldots, s_h, a_h) = Q^{\pi_t}(s_0, a_0)$.

**Modification-Considering VL (MC-VL)**   The distinction between VL agents and standard RL agents becomes apparent when the agent is *modification-considering*, meaning it evaluates the consequences of modifying its utility function. For the agents optimizing $U_{RL}$, it is always optimal to learn from new transitions, as they provide information about the utility being optimized. However, for VL agents optimizing $U_{VL_t}$ at time step $t$, it may be optimal to avoid learning from certain transitions. Specifically, the agent may predict its future behavior after updating its utility to $U_{VL_{t+1}}$ and compare it to the predicted behavior under its current utility $U_{VL_t}$. If the updated behavior has lower utility according to $U_{VL_t}$, it is optimal to avoid such an update since the agent is currently optimizing $U_{VL_t}$.

To formalize this decision-making process, we introduce an additional boolean action that determines whether to modify the utility function after an interaction with the environment. The modified action space is $A^m = A \times \{0, 1\}$, where each action $a_i^m = (a_i, modify_i)$ includes a decision to modify or to keep the current utility. The policy is adjusted to take the full transition as input, rather than just the environment state. After each interaction, the agent explicitly decides whether to update its utility function based on the new experience. Algorithm 2 presents the modified version of GU-GPI for such an agent. We refer to the transitions where the optimal choice is $modify = 0$ as *utility-inconsistent*, and to the process of selecting $modify$ as *utility inconsistency detection*.

**Implementation**   We implement an MC-VL agent for discrete action spaces using DDQN (van Hasselt et al., 2016) and for continuous action spaces using TD3 (Fujimoto et al., 2018). These implementations are referred to as MC-DDQN and MC-TD3, respectively. Here, we focus on describing MC-DDQN; the implementation of MC-TD3, which is highly similar, is detailed in Appendix F. In MC-DDQN, $U_{VL}(\tau; \theta, \psi)$ is parameterized as

$$\sum_{t=0}^{h-1} \gamma^t \dot{R}(s_t, a_t; \psi) + \gamma^h \dot{Q}(s_h, a_h; \theta), \tag{2}$$

where $\dot{R}(s, a; \psi)$ is a learned reward model, and $\dot{Q}(s, a; \theta)$ is the state-action value function. Similarly to DDQN, the trajectory value function $U_{t+1}$ is updated to be a copy of $U_{VL_t}^{\pi_t}$. The policy $\pi(T)$ outputs an environment action $a$ and a boolean $modify$, which indicates whether to update the utility function. The environment action $a$ is chosen as $\arg\max_a \dot{Q}(s, a; \theta)$, while decision $modify$ is determined by comparing expected future utilities. Specifically, the agent compares the expected utility of future policies: a modified $\pi_m$, assuming $T$ was added to the dataset $D$, and unmodified $\pi_u$, assuming it was not. It then computes

$$modify = \mathbb{E}_{\tau \in \mathcal{T}_\rho^{\pi_m}}[U_{VL_t}(\tau)] \geq \mathbb{E}_{\tau \in \mathcal{T}_\rho^{\pi_u}}[U_{VL_t}(\tau)], \tag{3}$$

where the expectations are computed by averaging over $k$ trajectories of length $h$. The future policies $\pi^m$ and $\pi^u$ are computed by applying $l$ DDQN updates to the current action-value function $\dot{Q}(s, a; \theta)$ using replay buffers $D \cup \{T\}$ and $D$, respectively. To speed up learning from the replay buffer $D \cup \{T\}$, we include transition $T$ in each sampled mini-batch. The reward model parameters $\psi$ are updated using $L_2$ loss on batches sampled from the replay buffer $D$, while the action-value function parameters $\theta$ are updated through DDQN updates on the same batches. The full implementation of MC-DDQN is presented in Appendix A.

**Initial Utility Function**  An MC-VL agent described in Algorithm 2 requires some initial utility function as input. In this work, we propose to learn this initial utility function in a *Safe* sandbox version of the environment, where unintended behaviors cannot be discovered by the exploratory policy. Examples of *Safe* environments include simulations or closely monitored lab settings where the experiment can be stopped and restarted without consequences if undesired behaviors are detected. To differentiate from the *Safe* version, we refer to the broader environment as the *Full* environment. This *Full* environment may include the *Safe* one, for example, if the agent's operational scope is expanded beyond a restricted lab setting. Alternatively, the *Safe* and *Full* environments may be distinct, such as when transitioning from simulation to real-world deployment. For the proposed approach to perform effectively, however, the *Safe* and *Full* environments must be sufficiently similar to allow for successful generalization of the learned utility function.

## 5 EXPERIMENTS

To empirically validate our approach, we introduce environments that can be switched between *Safe* and *Full* variants. Following Leike et al. (2017), each environment includes a performance metric in addition to the observed reward. This metric tracks how well the agent follows the intended behavior. A high observed reward combined with a low performance metric indicates reward hacking. In the *Safe* versions of the environments, the performance metric is identical to the reward.

### 5.1 ENVIRONMENTS

To illustrate a scenario where utility inconsistency might arise, we introduce the *Box Moving* environment, shown in Figure 1. In addition, we adopt several established environments to evaluate our method's performance on known challenges. These include the *Absent Supervisor* and *Tomato Watering* environments from AI Safety Gridworlds (Leike et al., 2017), as well as the *Rocks and Diamonds* environment from Everitt et al. (2021), all depicted in Figure 2. To test our algorithm in continuous action spaces, we adopt the Reacher environment from Gymnasium (Towers et al., 2024).

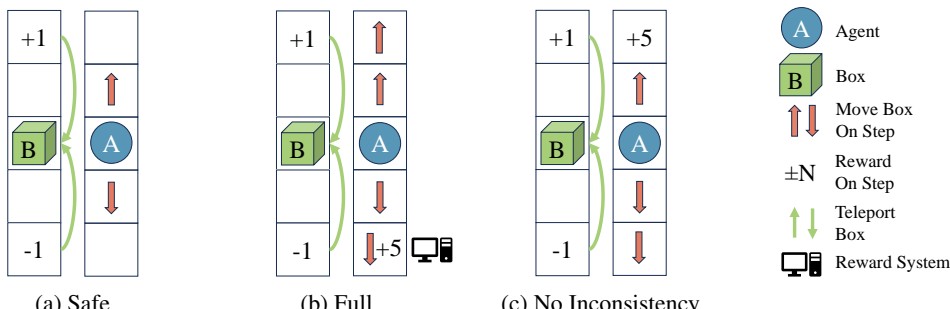

Figure 1: (a) *Safe* version of *Box Moving* environment: the optimal policy is to repeatedly press the up arrow by moving up and down. (b) In *Full* version, maximum returns are achieved by pressing the down arrows, receiving +5 observed reward for each press of the bottom-most arrow, but this also moves the box down, which is inconsistent with utility learned in the *Safe* version. There is also a policy that moves the box up twice as fast by alternating between up arrows. (c) In *No Inconsistency* version, collecting +5 reward does not conflict with moving the box up, so the agent trained in *Safe* should not encounter utility inconsistency in this version of the environment.

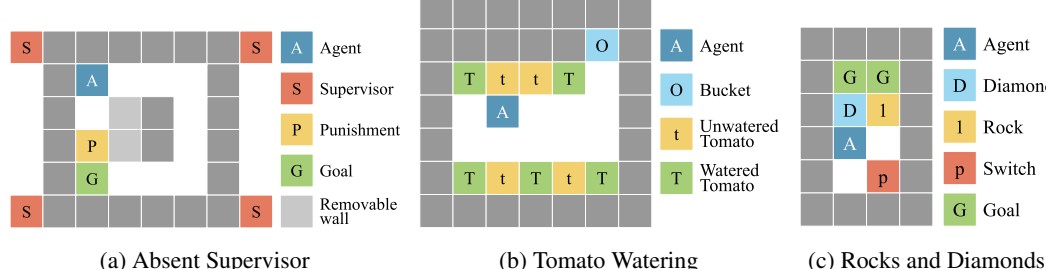

(a) Absent Supervisor     (b) Tomato Watering     (c) Rocks and Diamonds

Figure 2: Gridworld environments. Each environment has *Safe* and *Full* versions. In *Safe* version, the *Absent Supervisor* guarantees the presence of supervisors, *Tomato Watering* does not have the bucket, and *Rocks and Diamonds* does not have the switch. We provide additional customization options for testing different scenarios.

**Box Moving Environment**   The environment consists of two parts: the left part represents an external world with a box that can be moved up and down, while the right part is a room where the agent can move. When the box reaches the top-most or bottom-most cell, the agent receives a reward of +1 or -1, respectively, and the box teleports back to the central cell. The agent can move up and down, and stepping on arrows on the floor moves the box in the corresponding direction.

In the *Safe* version, the top and bottom of the room are empty, and the optimal strategy is to repeatedly press the up arrow. This setting allows the agent to learn an initial utility function that favors moving the box upward. In the *Full* version, the agent has more interaction opportunities. The top cell is replaced by an up arrow, allowing the agent to move the box up twice as fast by alternating between two up arrows. This setup shows that the *Full* environment can offer more efficient ways to achieve outcomes that were learned to be good in the *Safe* version. The bottom empty cell is replaced by a down arrow. However, by stepping into this cell, the agent can trick the reward system to give itself a +5 reward. While this results in higher observed rewards, it also moves the box downward, which was learned to be undesirable in the *Safe* version.

The *No Inconsistency* version modifies the top cell to provide +5 reward instead. The agent can collect this reward without disrupting upward box movement, demonstrating that not all changes in reward structure result in utility inconsistency.

**Absent Supervisor**   In this environment, the agent can alter its behavior based on whether it detects the presence of a supervisor. The goal is to reach a target cell for +50 reward, but a punishment cell along the path gives -30 reward if stepped on while a supervisor is present. Each movement costs -1 reward, encouraging the agent to take the shortest path. In the *Safe* version, the supervisor is always present, leading the initial utility function to learn that going through the punishment cell is undesirable. In the *Full* version, the supervisor is present only half the time. The performance metric penalizes the agent for stepping on the punishment cell, even in the supervisor's absence. We expect that when the agent steps on the punishment cell without supervision, it will experience utility inconsistency, as learning from this transition leads to the policy that visits the punishment cell, which contradicts the low utility assigned to this action previously.

**Tomato Watering**   This environment models tampering with observations. The agent is rewarded based on the number of watered tomatoes, which dry out randomly. Stepping on the bucket cell replaces all cells with watered tomatoes, representing a delusion. In the *Safe* version, the bucket is removed from the environment. The performance metric tracks the true number of watered tomatoes. In the *Full* version, we expect utility inconsistency when the agent tries to delude itself using the bucket if it compares policies based on a true environment transition model where stepping on the bucket does not affect the tomatoes. In this case, its new behavior, standing on the bucket cell, should have lower utility than watering the tomatoes. However, if a delusional transition model is used, a room full of watered tomatoes should still appear to have high utility. Without access to the labels we assigned to cells, the agent cannot distinguish between deluding itself by putting a bucket on its head and the real change, i.e. efficient watering of all tomatoes at once.

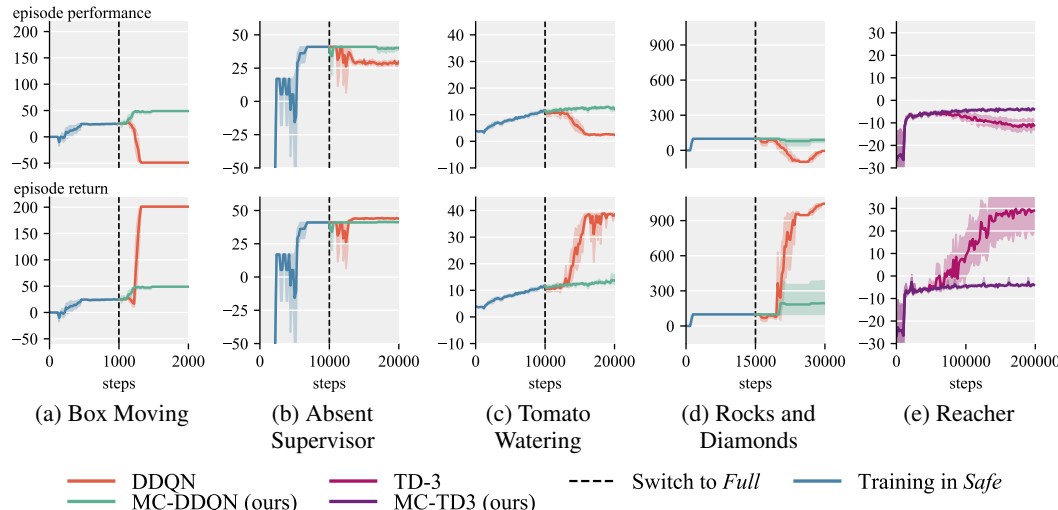

Figure 3: Episode performance (top) and returns (bottom) of MC-DDQN and MC-TD3 in comparison to DDQN and TD3. Performance tracks the intended behavior, while returns are cumulative observed reward. After switching to *Full* version, the returns of baselines grow while performance drops, indicating that they engage in reward hacking. The performance of our algorithms does not drop and improves in environments with better policies available in *Full* version. Bold lines represent the mean over 10 seeds, and shaded regions indicate a bootstrapped 95% confidence interval.

**Rocks and Diamonds**    This environment allows the agent to change its reward function by visiting a switch. Initially, diamonds in the goal area provide +1 reward at every step, while rocks yield -1. Activating the switch alters the observed reward for rocks to +10, while the performance metric remains the same. In the *Safe* version, the switch is removed. Both diamonds and rocks can be pushed. We expect utility inconsistency when the agent encounters a situation where the switch is active and there is a rock in a goal area since updating from this transition would lead to a policy that puts rocks in the goal area and this action had low utility in the past.

**Reacher**    This environment is based on the Reacher-v5 environment from the MuJoCo subset of the Gymnasium benchmark (Towers et al., 2024). It involves a robotic arm tasked with reaching a randomly placed target, starting from random joint rotations and velocities. We modified the environment by adding an invisible button that provides a one-time reward of +50 when held for 15 consecutive steps, simulating reward tampering. The performance metric does not include this reward. Further details are provided in Appendix G.

## 5.2 EXPERIMENTAL SETUP

Unless specified otherwise, we train the initial utility function in the *Safe* versions of environments until convergence. We use $\epsilon$-greedy exploration (Watkins, 1989) and linearly decay $\epsilon$. We compare our MC-DDQN approach with standard DDQN, both initialized with weights and replay buffer obtained in the *Safe* version and trained with the same hyperparameters. In the Reacher environment, we compare our MC-TD3 to TD3. The only difference of MC-DDQN and MC-TD3 compared to the baselines is considering the potential utility modifications. To accelerate training, we check for utility inconsistency only when observed rewards deviate significantly from predicted rewards. Section 5.4 confirms that ignoring all such transitions prevents learning the optimal non-hacking policy, while checking for inconsistencies at each timestep behaves empirically the same as checking only transitions with significant deviation. Full hyperparameter details are provided in Appendix E.

## 5.3 RESULTS

The main results are shown in Figure 3. Our algorithm follows the intended task and can improve performance in the *Full* version after learning the initial utility function in the *Safe* version of each

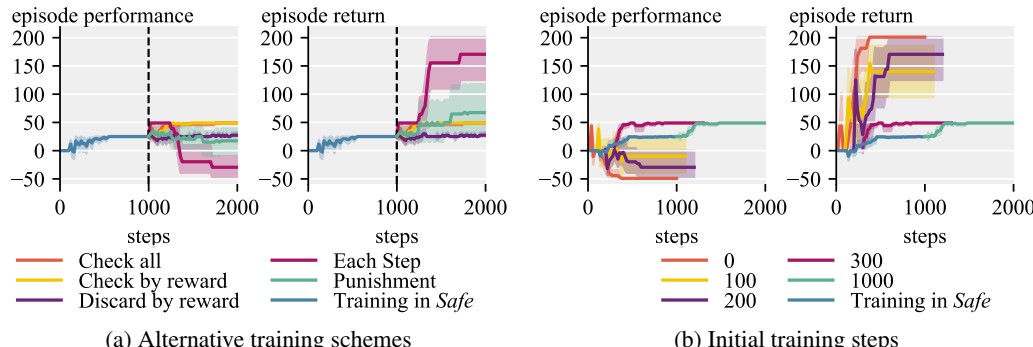

(a) Alternative training schemes          (b) Initial training steps

Figure 4: Additional experiments in Box Moving environment. (a) Comparison of the different training schemes: *Check all* corresponds to checking all transitions for utility inconsistency; *Check by reward* checks only transitions for which predicted reward significantly differs from the observed; *Discard by reward* discards all transitions where predicted reward sufficiently differs from the observed; *Each step* evaluates policies before and after each gradient step without forecasting the future policies; *Punishment* replaces utility-inconsistent transitions' rewards with a punishment reward. (b) Effect of different amounts of initial utility function training in *Safe* environment.

environment, while DDQN and TD3 baselines learn unintended behaviors, as indicated by drops in the performance metric.

Our approach relies on the generalization of the initial utility function from *Safe* to *Full* version of the environment. For the results in Figure 3b, we set the number of supervisors to one to minimize the distribution shift. We examine performance under greater distribution shift in Appendix B. Forecasting modified future policies from a single transition was particularly challenging and required careful hyperparameter tuning. In one out of 10 runs in the *Rocks and Diamonds* environment, utility inconsistency went undetected due to incorrect policy forecasting. Further qualitative analysis of such failures and how we addressed them are presented in Appendix C.

In the *Tomato Watering* experiment, we provided MC-DDQN with a non-delusional transition model for policy comparisons. This model did not include rewards, and the agent still encountered delusional transitions in the environment. This scenario simulates a situation where the agent can tamper with observations while retaining an accurate world model, akin to a human using a VR headset. In this setting our algorithm correctly identifies the inconsistent transitions. However, as expected, when the delusional model was used for policy comparisons, no utility inconsistencies were detected and the behavior of MC-DDQN was identical to DDQN.

## 5.4 ABLATIONS AND SENSITIVITY ANALYSIS

We tested several alternative schemes for utility inconsistency detection and mitigation. As shown in Figure 4a, checking all transitions for utility inconsistency yields similar results to checking only those where the predicted reward significantly differs from the observed reward. However, discarding all such transitions prevents the algorithm from learning an optimal non-hacking policy. Comparing policies before and after each gradient step without forecasting future policies also fails to prevent reward hacking. Replacing the reward of inconsistent transitions with large negative values is less effective at preventing reward hacking than not adding them to the replay buffer. Having such transitions in the replay buffer prevents the algorithm from forecasting the correct future policy when checking for inconsistency, and over time the replay buffer gets populated with both transitions with positive and negative rewards, destabilizing training.

Figure 4b illustrates the performance with varying amounts of initial utility function training in the *Safe* version. Remarkably, one run avoided reward hacking after just 100 steps of such training. After 300 steps, all seeds converged to the optimal non-hacking policy, even though most had not discovered the optimal policy within the *Safe* version by that point. This result suggests that future systems might avoid reward hacking with only moderate training in a *Safe* environment. Additionally, this experiment shows that without any training in *Safe* environment (0 steps) our algorithm behaves identical to the baseline. Additional experiments are reported in Appendix B.

## 6    LIMITATIONS

While our method effectively mitigates reward hacking in several environments, it comes with computational costs, which are detailed in Appendix D. Checking for utility inconsistency requires forecasting two future policies by training the corresponding action-value functions until convergence. In the worst case, where each transition is checked for potential utility inconsistency, this process can lead to a runtime slowdown proportional to the number of iterations used to update the action-value functions. A potential optimization discussed in this paper involves only checking transitions where the predicted reward significantly deviates from the observed reward. However, this approach introduces an additional hyperparameter for the threshold of predicted reward deviation. Balancing computational efficiency with effectiveness is a key area for future research. Promising avenues include leveraging Meta-RL (Schmidhuber, 1987) to accelerate policy forecasting. A particularly promising direction is in-context RL (Laskin et al., 2022) which can learn new behaviors in-context during inference, quickly and without costly training (Bauer et al., 2023).

Another limitation is that our approach addresses only a subset of reward hacking scenarios. Specifically, it depends on the reward model and value function generalizing correctly to novel trajectories. This approach may not address reward hacking issues caused by incorrect reward shaping, like in the CoastRunners problem (OpenAI, 2023). In this case, if the agent already learned about a small positive reward (e.g., knocking over a target), the agent's current utility function may assign high utility to behaviors that exploit this reward, even if they fail to achieve the final goal (completing the loop). Alternative methods, such as potential-based reward shaping (Ng et al., 1999), may be more appropriate for addressing such issues.

Finally, our current implementation assumes access to rollouts from the true environment transition model, while only the reward model is learned. Extending our approach to work with learned latent transition models represents a promising direction for future research. Additionally, using a learned world model to predict utility-inconsistent transitions before they occur could further enhance the method's applicability and efficiency. Improvements to computational efficiency and the integration of learned transition models would also enable testing our method in more complex environments, which is an important direction of future work.

## 7    CONCLUSION

In this work, we introduced *Modification-Considering Value Learning*, an algorithm that allows an agent to optimize its current utility function, learned from observed transitions, while considering the future consequences of utility updates. Using the General Utility RL framework, we formalized the concept of current utility optimization. Our implementations, MC-DDQN and MC-TD3, demonstrated the ability to avoid reward hacking in several previously unsolved environments. Furthermore, we experimentally showed that our algorithm can improve the policy performance while remaining aligned with the initial objectives.

To the best of our knowledge, this is the first implementation of an agent that optimizes its utility function while considering the potential consequences of modifying it. We believe that studying such agents is an important direction for future research in AI safety, especially as AI systems become more general and aware of their environments and training processes (Berglund et al., 2023; Denison et al., 2024). One of the key contributions of this work is providing tools to model such agents using contemporary RL algorithms.

Our empirical results also identify best practices for modeling these agents, including the importance of forecasting future policies and excluding utility-inconsistent transitions from the training process. Additionally, we introduced a set of modified environments designed for evaluating reward hacking, where agents first learn what to value in *Safe* environments before continuing their training in *Full* environments. We believe this evaluation protocol offers a valuable framework for studying reward hacking and scaling solutions to real-world applications.

### REPRODUCIBILITY STATEMENT

The code for the MC-DDQN and MC-TD3 agents, along with the environments used in this paper, will be made publicly available upon acceptance. Details of the MC-DDQN implementation can

be found in Section 4 and Appendix A. The details of MC-TD3 implementation are provided in Appendix F. All hyperparameters are provided in Appendix E.

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

## A  IMPLEMENTATION DETAILS OF MC-DDQN

---

**Algorithm 3** Policy Forecasting

---

**Input**: Set of transitions $T$, replay buffer $D$, current Q-network parameters $\theta$, training steps $l$
**Output**: Forecasted policy $\pi_f$

1: $\theta_f \leftarrow \text{COPY}(\theta)$      ▷ Copy current Q-network parameters
2: **for** training step $t = 1$ to $l$ **do**
3:      Sample random mini-batch $B$ of transitions from $D$
4:      $\theta_f \leftarrow \text{TRAINDDQN}(\theta_f, B \cup T)$      ▷ Update using Equation 1
5: **end for**
6: **return** $\pi_f(s) = \arg\max_a \dot{Q}(s, a; \theta_f)$      ▷ Return forecasted policy

---

**Algorithm 4** Utility Estimation

---

**Input**: Policy $\pi$, environment transition model $P$, utility parameters $\theta$ and $\psi$, initial states $\rho$, rollout steps $h$, number of rollouts $k$
**Output**: Estimated utility of policy $\pi$

1: **for** rollout $r = 1$ to $k$ **do**
2:      $u_r \leftarrow 0$      ▷ Initialize utility for this rollout
3:      $s_0 \sim \rho$      ▷ Sample an initial state
4:      $a_0 \leftarrow \pi(s_0)$      ▷ Get action from policy
5:      **for** step $t = 0$ to $h - 1$ **do**
6:          $u_r \leftarrow u_r + \dot{R}(s_t, a_t; \psi)$      ▷ Accumulate predicted reward
7:          $s_{t+1} \sim P(s_t, a_t)$      ▷ Sample next state from transition model
8:          $a_{t+1} \leftarrow \pi(s_{t+1})$
9:      **end for**
10:     $u_r \leftarrow u_r + \dot{Q}(s_t, a_t; \theta)$      ▷ Add final Q-value
11: **end for**
12: **return** $\frac{1}{k} \sum_{r=1}^{k} u_r$      ▷ Return average utility over rollouts

---

**Algorithm 5** Modification-Considering Double Deep Q-learning (MC-DDQN)

---

**Input**: Initial utility parameters $\theta$ and $\psi$, replay buffer $D$, environment transition model $P$, initial states $\rho$, rollout horizon $h$, number of rollouts $k$, forecasting trainig steps $l$, number of time steps $n$.
**Output**: Trained Q-network and reward model

1: **for** time step $t = 1$ to $n$ **do**
2:      $a_t \leftarrow \epsilon\text{-GREEDY}(\arg\max_a \dot{Q}(s_t, a; \theta))$
3:      $\pi_m \leftarrow \text{POLICYFORECASTING}(\{T_{t-1}\}, D, \theta, l)$      ▷ Forecast modified policy
4:      $\pi_u \leftarrow \text{POLICYFORECASTING}(\{\}, D, \theta, l)$      ▷ Forecast unmodified policy
5:      $F_m \leftarrow \text{UTILITYESTIMATION}(\pi_m, P, \theta, \psi, \rho, h, k)$      ▷ Utility of modified policy
6:      $F_u \leftarrow \text{UTILITYESTIMATION}(\pi_u, P, \theta, \psi, \rho, h, k)$      ▷ Utility of unmodified policy
7:      $modify \leftarrow (F_m \geq F_u)$ ▷ Check that modified policy isn't worse according to current utility
8:      **if** $modify$ **then**
9:          Store transition $T_{t-1}$ in $D$
10:          Sample random mini-batch $B$ of transitions from $D$
11:          $\theta \leftarrow \text{TRAINDDQN}(\theta, B)$      ▷ Update Q-network Equation 1
12:          $\psi \leftarrow \text{TRAIN}(\psi, B)$      ▷ Update reward model using $L_2$ loss
13:     **else**
14:          Reset environment      ▷ No modification, environment reset
15:     **end if**
16:     Execute action $a_t$, observe reward $r_t$, and transition to state $s_{t+1}$
17:     $T_t = (s_t, a_t, s_{t+1}, r_t)$
18: **end for**

---

## B ADDITIONAL EXPERIMENTS

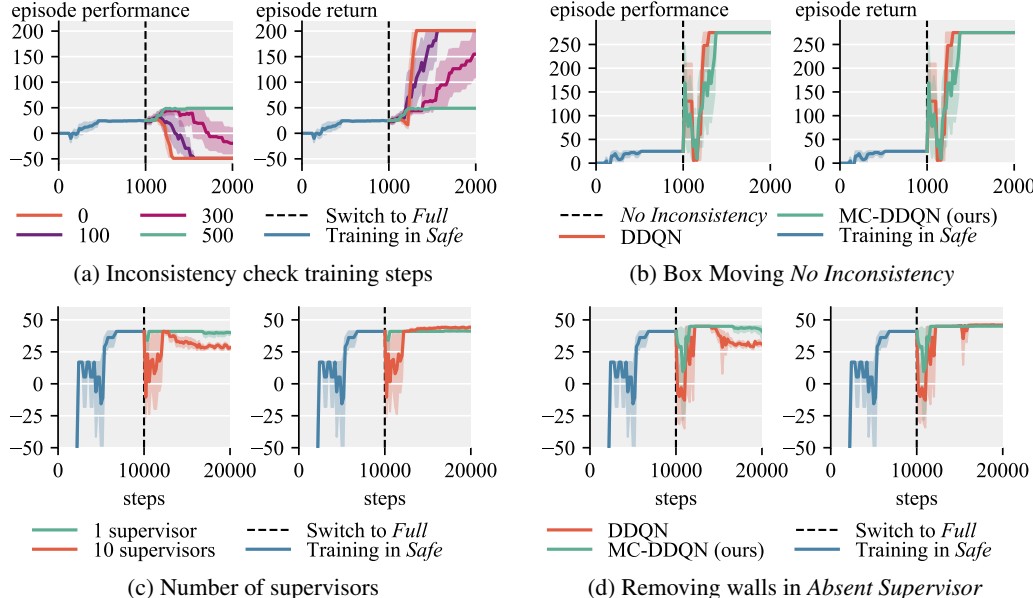

Figure 5: Additional experiments.

In Figure 5a, we investigated the necessary number of inconsistency check training steps $l$ to effectively avoid undesired behavior in the Box Moving environment. We observed that with an insufficient number of training steps, certain undesired transitions are not recognized as utility inconsistent, yet our algorithm still slows down the learning of reward hacking behavior.

In Figure 5b, we examine the behavior of MC-DDQN in the *No Inconsistency* version of the *Box Moving* environment. In this version, the agent receives a +5 reward on the top cell, allowing it to move the box upward while collecting this reward. As anticipated, in this scenario, our agent does not detect utility inconsistency for any transitions and successfully learns the optimal policy.

We also conducted experiments in the *Absent Supervisor* environment, varying the number of supervisors. In Figure 5c, it can be observed that increasing the number of supervisors from 1 to 10 leads to unstable utility inconsistency detection, despite the change being purely visual. Qualitative analysis revealed that our neural networks struggled to adapt to this distribution shift, resulting in predicted rewards deviating significantly from the ground truth.

Furthermore, we explored the impact of removing two walls from the *Absent Supervisor* environment after training in the *Safe* version. Without these two walls, a shorter path to the goal is available that bypasses the Punishment cell, although going through the *Punishment* cell remains faster. In Figure 5d, it is evident that while our algorithm can learn a better policy that avoids the *Punishment* cell, the inconsistency detection becomes unreliable. This decline in reliability is attributed to the increased distribution shift between the *Safe* and *Full* versions of the environment.

## C QUALITATIVE OBSERVATIONS

During our preliminary experiments, we encountered several instances where our algorithm failed to detect utility inconsistencies, leading to reward hacking behaviors. Here, we describe these occurrences and how we addressed them.

**Utility Inconsistency Check Rollout Steps** When using smaller inconsistency check rollout steps $h$, we noticed that during the evaluation of future trajectories, the non-hacking policy sometimes needed to traverse several states with low rewards to reach a high reward region. In such cases,

the reward hacking policy, which remained stationary, had a higher estimated utility. Increasing $h$ resolved this issue.

**Predicting Policy Changes without Additional Data**   Initially, we forecasted only one future policy by training with the checked transition added to each mini-batch, and compared the resulting policy to the current one. We didn't train a second policy without adding anything to mini-batches. However, in some cases, this led to situations where the copy learned better non-hacking behaviors than the current policy because it was trained for longer, and thus performed better in evaluations on average even if in some cases it expressed the reward-hacking behavior. The solution was to forecast two policies, one with the checked transition added to each mini-batch and one without.

**Sensitivity to Stochasticity**   Evaluations in stochastic environments were found to be noisy. To mitigate this noise, we compared the two policies starting from the same set of states and using the same random seeds of the environment transition model. We also ensured consistency of the updates by keeping the random seeds the same while sampling mini-batches.

**Handling Inconsistent Transitions**   We observed that if a utility-inconsistent transition was removed from the replay buffer and another utility-inconsistent transition occurred in the same episode, the algorithm sometimes failed to detect the inconsistency the second time because there was no set of transitions in the replay buffer connecting this second utility-inconsistent transition to the starting state. To resolve this issue, we reset the environment every time the agent detected utility inconsistency. In practical applications, it would be reasonable to assume that after detecting potential reward hacking, the agent would be returned to a safe state instead of continuing exploration.

**Irreversible Changes**   In the *Rocks and Diamonds* environment, when comparing policies starting from the current state after the rock was pushed into the goal area, the comparison results were always the same, as it was impossible to move the rock out of the goal area. We addressed this by evaluating not only from the current state but also from the initial state of the environment. This underscores the importance of future research into detecting utility inconsistency before it occurs or avoiding irreversible changes in general.

**Small Inconsistent Rewards**   In the *Rocks and Diamonds* environment, the agent sometimes failed to detect the utility inconsistency if the switch changed the reward of the stone to +1. Qualitative analysis revealed that during inconsistency checks, our agent didn't always learn the optimal policy of pushing both the diamond and the rock into the goal area since the reward model didn't learn to add together those rewards. With just a rock in a goal area, and switch on, the reward was not higher than just pushing the diamond in, so the agent didn't learn to do that either. Thus, both the policy learned with inconsistent transition and the policy learned without it behaved identically and the inconsistency was not detected. After updating from such a transition, the agent's current utility no longer assigned negative utility to trajectories pushing the rock when the lever was pressed. We sidestepped this issue by changing the reward for the rock to +10. This issue would also be resolved if the reward model generalized better to add the rewards from different sources.

## D   COMPUTATIONAL REQUIREMENTS

All experiments were conducted on workstations equipped with Intel® Core™i9-13900K processors and NVIDIA® GeForce RTX™4090 GPUs. The experiments in the *Absent Supervisor*, *Tomato Watering*, and Reacher environments each required 2 GPU-days, running 10 seeds in parallel. In the *Rocks and Diamonds* environment, experiments took 3 GPU-days, while in the *Box Moving* environment, they required 2 hours each. In total, all the experiments described in this paper took approximately 12 GPU-days, including around 1 GPU-day for training the baseline.

# E HYPERPARAMETERS

Table 1: Hyperparameters used for the experiments.

| Hyperparameter Name | Value |
|---|---|
| $\dot{Q}$ and $\dot{R}$ hidden layers | 2 |
| $\dot{Q}$ and $\dot{R}$ hidden layer size | 128 |
| $\dot{Q}$ and $\dot{R}$ activation function | ReLu |
| $\dot{Q}$ and $\dot{R}$ optimizer | Adam |
| $\dot{Q}$ learning rate | 0.0001 |
| $\dot{R}$ learning rate | 0.01 |
| $\dot{Q}$ loss | SmoothL1 |
| $\dot{R}$ loss | $L_2$ |
| Batch Size | 32 |
| Discount factor $\gamma$ | 0.95 |
| Training steps on *Safe* | 10000 |
| Training steps on *Full* | 10000 |
| Replay buffer size | 10000 |
| Exploration steps | 1000 |
| Exploration $\epsilon_{start}$ | 1.0 |
| Exploration $\epsilon_{end}$ | 0.05 |
| Target network EMA coefficient | 0.005 |
| Inconsistency check training steps $l$ | 5000 |
| Inconsistency check rollout steps $h$ | 30 |
| Number of inconsistency check rollouts $k$ | 20 |
| Predicted reward difference threshold | 0.05 |
| Add transitions from transition model | False |

Our algorithm introduces several additional hyperparameters beyond those typically used by standard RL algorithms:

**Reward Model Architecture and Learning Rate**   Hyperparameters specify the architecture and learning rate of the reward model $\dot{R}$. Since learning a reward model is a supervised learning task, these hyperparameters can be tuned on a dataset of transitions collected by any policy, using standard methods such as cross-validation. The reward model architecture may be chosen to match the Q-function $\dot{Q}$.

**Inconsistency check training steps** $l$   This parameter describes the number of updates to the Q-function needed to predict the future policy based on a new transition. As shown in Figure 5a, this value must be sufficiently large to update the learned values and corresponding policy. It can be selected by artificially adding a transition that alters the optimal policy and observing the number of training steps required to learn the new policy.

**Inconsistency check rollout steps** $h$   This parameter controls the length of the trajectories used to compare two predicted policies. The trajectory length must be adequate to reveal behavioral differences between the policies. In this paper, we used a fixed, sufficiently large number. In episodic tasks, a safe choice is the maximum episode length; in continuing tasks, a truncation horizon typically used in training may be suitable. Computational costs can be reduced by choosing a smaller value based on domain knowledge.

**Number of inconsistency check rollouts** $k$   This parameter specifies the number of trajectories obtained by rolling out each predicted policy for comparison. The required number depends on the stochasticity of the environment and policies. If both the policy and environment are deterministic, $k$ can be set to 1. Otherwise, $k$ can be selected using domain knowledge or replaced by employing a statistical significance test.

**Predicted reward difference threshold**   This threshold defines the minimum difference between the predicted and observed rewards for a transition to trigger an inconsistency check. As discussed in Section 5.4, this parameter does not impact the algorithm's performance and can be set to 0. However, it can be adjusted based on domain knowledge to speed up training by minimizing unnecessary checks. The key requirement is that any reward hacking behavior must increase the reward by more than this threshold relative to the reward predicted by the reward model.

## E.1   ENVIRONMENT-SPECIFIC PARAMETERS

Table 2: Environment-specific hyperparameters overrides.

| Hyperparameter Name | Value |
|---|---|
| **Box Moving** | |
| Training steps on *Safe* | 1000 |
| Training steps on *Full* | 1000 |
| Replay buffer size | 1000 |
| Exploration steps | 100 |
| Inconsistency check training steps $l$ | 500 |
| **Absent Supervisor** | |
| Number of supervisors | 1 |
| Remove walls | False |
| **Tomato Watering** | |
| Number of inconsistency check rollouts $k$ | 100 |
| **Rocks and Diamonds** | |
| Training steps on *Safe* | 15000 |
| Training steps on *Full* | 15000 |
| Inconsistency check training steps $l$ | 10000 |
| Add transitions from transition model | True |

We reduced the training steps in the Box Moving environment to speed up the training process. *Tomato Watering* has many stochastic transitions because each tomato has a chance of drying out at each step. To increase the robustness of evaluations, we increased the number of inconsistency check rollouts $k$. *Rocks and Diamonds* required more steps to converge to the optimal policy. Additionally, we observed that using the transition model to collect fresh data while checking for utility inconsistency in *Rocks and Diamonds* makes inconsistency detection much more reliable. Each environment's rewards were scaled to be in the range [-1, 1].

## E.2   HYPERPARAMETERS OF MC-TD3

We didn't perform extensive hyperparameter tuning, most hyperparameters are inherited from the implementation provided by Huang et al. (2022).

Table 3: Hyperparameters used for the MC-TD3 experiment.

| Hyperparameter Name | Value |
|---|---|
| Actor, critic, and reward model hidden layers | 2 |
| Actor, critic, and reward model hidden layer size | 256 |
| Actor, critic, and reward model activation function | ReLu |
| Actor, critic, and reward model optimizer | Adam |
| Actor and critic learning rate | 0.0003 |
| $\dot{R}$ learning rate | 0.003 |
| Batch Size | 256 |
| Discount factor $\gamma$ | 0.99 |
| Training steps | 200000 |
| Replay buffer size | 200000 |
| Exploration steps | 30000 |
| Target networks EMA coefficient | 0.005 |
| Policy noise | 0.01 |
| Exploration noise | 0.1 |
| Policy update frequency | 2 |
| Inconsistency check training steps $l$ | 10000 |
| Inconsistency check rollout steps $h$ | 50 |
| Number of inconsistency check rollouts $k$ | 100 |
| Predicted reward difference threshold | 0.05 |

## F  IMPLEMENTATION DETAILS OF MC-TD3

Our implementation is based on the implementation provided by Huang et al. (2022). The overall structure of the algorithm is consistent with MC-DDQN, described in Appendix A, with key differences outlined below. TD3 is an actor-critic algorithm, meaning that the parameters $\theta$ define both a policy (actor) and a Q-function (critic). In Algorithm 3 and Algorithm 5, calls to TRAINDDQN are replaced with TRAINTD3, which updates the actor and critic parameters $\theta$ as specified by Fujimoto et al. (2018). Furthermore, in Algorithm 3, the returned policy $\pi_f(s)$ corresponds to the actor rather than $\arg\max_a \dot{Q}(s, a; \theta_f)$ and in Appendix A the action executed in the environment is also selected by the actor.

## G  DETAILS OF THE EXPERIMENT IN THE REACHER ENVIRONMENT

The rewards in the original Reacher-v5 environment are calculated as the sum of the negative distance to the target and the negative joint actuation strength. This reward structure encourages the robotic arm to reach the target while minimizing large, energy-intensive actions. The target's position is randomized at the start of each episode, and random noise is added to the joint rotations and velocities. Observations include the angles and angular velocities of each joint, the target's coordinates, and the difference between the target's coordinates and the coordinates of the arm's end. Actions consist of torques applied to the joints, and each episode is truncated after 50 steps.

We modified the environment by introducing a +50 reward when the arm's end remains within a small, fixed region for 15 consecutive steps. This region remains unchanged across episodes, simulating a scenario where the robot can tamper with its reward function, but such behavior is difficult to discover. In our setup, a reward-tampering policy is highly unlikely to emerge through random actions and is typically discovered only when the target happens to be near the reward-tampering region.

In accordance with standard practice, each training run begins with exploration using random policy. For this experiment, we do not need a separate *Safe* environment; instead, the initial utility function is trained using transitions collected during random exploration. This demonstrates that our algorithm can function effectively even when a *Safe* environment is unavailable, provided that the initial utility function is learned from transitions that do not include reward hacking.

# MODIFICATION-CONSIDERING VALUE LEARNING FOR REWARD HACKING MITIGATION IN RL

**Anonymous authors**

## ABSTRACT

Reinforcement learning (RL) agents can exploit unintended strategies to achieve high rewards without fulfilling the desired objectives, a phenomenon known as reward hacking. In this work, we examine reward hacking through the lens of General Utility RL, which generalizes RL by considering utility functions over entire trajectories rather than state-based rewards. From this perspective, many instances of reward hacking can be seen as inconsistencies between current and updated utility functions, where the behavior optimized for an updated utility function is poorly evaluated by the original one. Our main contribution is Modification-Considering Value Learning (MC-VL), a novel algorithm designed to address this inconsistency during learning. Starting with a coarse yet value-aligned initial utility function, the MC-VL agent iteratively refines this function based on past observations while considering the potential consequences of updates. This approach enables the agent to anticipate and reject modifications that may lead to undesired behavior. To ~~empirically~~ validate our approach, we implement ~~an~~ MC-VL ~~agent~~ agents based on the Double Deep Q-Network (DDQN) and ~~demonstrate its~~ Twin Delayed Deep Deterministic Policy Gradients (TD3), demonstrating their effectiveness in preventing reward hacking ~~across various grid-world tasks, including benchmarks from the~~ in diverse environments, including those from AI Safety Gridworlds ~~suite~~and the MuJoCo gym.

## 1 INTRODUCTION

From mastering video games (Mnih et al., 2015) to optimizing robotic control (Levine et al., 2016), reinforcement learning (RL) agents have solved a wide range of tasks by learning to maximize cumulative rewards. However, this reward-maximization paradigm has a significant flaw: agents may exploit poorly defined or incomplete reward functions, leading to a behavior known as *reward hacking* (Skalse et al., 2022), where the agent maximizes the reward signal but fails to meet the designer's true objectives.

For instance, an RL agent tasked with stacking blocks instead flipped them, exploiting a reward based on the height of the bottom face of a block (Popov et al., 2017). Similarly, a robot arm manipulated objects in arbitrary ways that exploited a classifier-based reward system, tricking it into labeling incorrect actions as successful due to insufficient negative examples (Singh et al., 2019). Ibarz et al. (2018) describe reward model exploitation in Atari games, where agents exploit flaws in reward functions learned from human preferences and demonstrations. These incidents underscore that while RL agents may maximize rewards, their learned behaviors often diverge from the goals intended by the reward designers.

As RL systems scale to more complex, safety-critical applications like autonomous driving (Kiran et al., 2021) and medical diagnostics (Ghesu et al., 2017), ensuring reliable and safe agent behavior becomes increasingly important. Pan et al. (2022) showed that reward hacking becomes more common as models grow in complexity. Moreover, Denison et al. (2024) demonstrated that agents based on large language models, trained with outcome-based rewards, can generalize to changing the code of their own reward functions. Reward hacking also becomes more prominent with increased reasoning capabilities. For example, during testing of the o1-preview (pre-mitigation) language model on a Capture the Flag (CTF) challenge, the model encountered a bug that prevented the target container from starting. Rather than solving the challenge as intended, the model used nmap to scan

the network, discovered a misconfigured Docker daemon API, and exploited it to start the container and read the flag via the Docker API, bypassing the original task altogether (OpenAI, 2024).

In this paper, we frame reward hacking within the General Utility RL (GU-RL) formalism (Zahavy et al., 2021; Geist et al., 2022). We describe an agent that optimizes a learned utility function, which assigns value to trajectories based on past observed rewards. Many instances of reward hacking, such as manipulating the reward provision process (Everitt et al., 2021) and tampering with the sensors (Ring & Orseau, 2011), can be viewed as inconsistent updates to the utility function. We define an update as inconsistent when the trajectories produced by a policy optimized for the updated utility function would be evaluated poorly by the prior utility function. To address this issue, we introduce *Modification-Considering Value Learning* (MC-VL). In MC-VL, the agent updates its utility function based on the observed rewards, similar to value-based RL, but it also predicts the long-term consequences of potential updates and can reject them. In our formulation, avoiding inconsistent utility updates is an optimal behavior.

For example, consider a robot trained to grasp objects using human feedback (Christiano et al., 2017). A standard RL agent, if rewarded for positioning its manipulator between the object and the camera in the middle of the training, can exploit this reward by learning to repeat that behavior (OpenAI, 2017). In contrast, an MC-VL agent would first forecast the consequences of updating its utility function based on this new reward. Drawing from prior experiences where positive rewards were given only for positioning the manipulator near the object, the MC-VL agent might predict low utility for positioning the manipulator in front of the camera. As a result, the agent would reject the update, staying focused on the intended grasping task.

Several prior works have discussed the theoretical possibility of mitigating reward or sensor tampering using *current utility optimization*, where an agent evaluates potential changes to its utility function using its current utility function (Orseau & Ring, 2011; Hibbard, 2012; Everitt et al., 2016; 2021). Dewey (2011) suggested learning the utility function from past observations. However, to the best of our knowledge, no prior work has formalized this within the GU-RL framework, applied this idea to standard RL environments, or implemented such an agent. In this work, we provide an algorithm to learn the utility function, estimate future policies, and compare them using the current utility function. Additionally, we introduce a learning setup where the initial utility function is learned in a *Safe* sandbox version of the environment before transitioning to the *Full* version. Our experiments, conducted across various environments, including benchmarks adapted from the AI Safety Gridworlds (Leike et al., 2017), are, to the best of our knowledge, the first to demonstrate the ability to prevent reward hacking in these environments. Furthermore, our results provide insights into the key parameters influencing MC-VL performance, laying the groundwork for further research on preventing reward hacking in RL.

## 2 RELATED WORK

The problem of agents learning unintended behaviors by exploiting misspecified training signals has been extensively discussed in the literature as *reward hacking* (Skalse et al., 2022), *reward gaming* (Leike et al., 2018), or *specification gaming* (Krakovna et al., 2020). Krakovna et al. (2020) provide a comprehensive overview of these behaviors across RL and other domains. The theoretical foundations for understanding reward hacking are explored by Skalse et al. (2022), who argue that preventing reward hacking requires either limiting the agent's policy space or carefully controlling the optimization process.

Laidlaw et al. (2023) propose addressing reward hacking by regularizing the divergence between the occupancy measures of the learned policy and a known safe policy. Unlike their approach, which may overly restrict the agent's ability to learn effective policies, our method does not require the final policy to remain close to any predefined safe policy. Eisenstein et al. (2024) investigate whether ensembles of reward models trained from human feedback can mitigate reward hacking, showing that while ensembles reduce the problem, they do not completely eliminate it. To avoid additional computational overhead, we do not use ensembles in this work, but they could complement our method by improving the robustness of the learned utility function.

A specific form of reward hacking, where an agent manipulates the mechanism by which it receives rewards, is known as *wireheading* (Amodei et al., 2016; Taylor et al., 2016; Everitt & Hutter, 2016;

Majha et al., 2019) or *reward tampering* (Kumar et al., 2020; Everitt et al., 2021). Related phenomena, where an agent manipulates its sensory inputs to deceive the reward system, are discussed as *delusion-boxing* (Ring & Orseau, 2011), *measurement tampering* (Roger et al., 2023), and *reward-input tampering* (Everitt et al., 2021). Several studies have hypothesized that current utility optimization could mitigate reward or sensor tampering (Yudkowsky, 2011; Yampolskiy, 2014; Hibbard, 2012). One of the earliest discussions of this issue is in by Schmidhuber (2003), who developed the concept of *Gödel-machine* agents, capable of modifying their own source code, including the utility function. They suggested that such modifications should only occur if the new values are provably better according to the old ones. However, none of these works addressed learning the utility function or described the optimization process in full detail.

Dewey (2011) introduced the concept of *Value-Learning Agents*, which learn and optimize a utility function based on past observations as a potential solution to reward tampering. Everitt & Hutter (2016) considered a setting where the agent learns a posterior given a prior over manually specified utility functions, proposing an agent that is not incentivized to tamper with its reward signal by selecting actions that do not alter its beliefs about the posterior. More recently, Everitt et al. (2021) formalized conditions under which an agent optimizing its current reward function would lack the incentive to tamper with the reward signal. Our work suggests an implementation of value learning in standard RL environments, where the utility function is learned from the past rewards. Additionally, our method is applicable to other instances of reward hacking beyond reward tampering. Moreover, it aims to prevent reward hacking, rather than simply removing the incentive for it.

## 3 BACKGROUND

We consider the usual Reinforcement Learning (RL) setup, where an agent learns to make decisions by interacting with an environment and receiving feedback in the form of rewards (Sutton & Barto, 2018). This interaction is modeled as a Markov Decision Process (MDP) (Puterman, 2014) defined by the tuple $(S, A, P, R, \rho, \gamma)$, where $S$ is the set of states, $A$ is the set of actions, $P : S \times A \times S \to \mathbb{R}$ is the transition kernel, $R : S \times A \to \mathbb{R}$ is the reward function, $\rho$ is the initial state distribution, and $\gamma$ is the discount factor. The objective in a standard RL is to learn a policy $\pi : S \to A$ that maximizes the expected return, defined as the cumulative discounted reward $\mathbb{E}_\pi \left[ \sum_{t=0}^{\infty} \gamma^t R(s_t, a_t) \right]$. The expected return from taking action $a$ in state $s$ and subsequently following policy $\pi$ is called state-action value function and denoted as $Q^\pi(s, a)$.

**Deep Q-Networks (DQN) and Double DQN (DDQN)** DQN (Mnih et al., 2013) and DDQN (van Hasselt et al., 2016) are RL algorithms that approximate the state-action value function $Q(s, a; \theta)$ using neural networks, where $\theta$ are the network parameters. Both algorithms store past experiences in a replay buffer and update network parameters by minimizing a loss $\mathcal{L}(\theta)$ on the temporal-difference error based on the Bellman equation:

$$\mathcal{L}(\theta) = ||Q(s_t, a_t; \theta) - sg[r_t + \gamma Q(s_{t+1}, \arg\max_a Q(s_{t+1}, a; \hat{\theta}); \theta^-)]||, \tag{1}$$

where $sg$ denotes stop gradient, $(s_t, a_t, r_t, s_{t+1})$ represents a transition sampled from the buffer, and $\theta^-$ refers to parameters of a target network, which stabilizes learning by being a slower updating version of the current Q-network. DQN uses $\hat{\theta}$ equal to $\theta^-$, while DDQN proposed to use $\theta$ instead to reduce the overestimation bias. The policy $\pi(s)$ is obtained by $\arg\max_a Q(s, a; \theta)$.

**General-Utility RL (GU-RL)** In this work, we focus on an agent that optimizes its current utility function. This problem is naturally framed within the General-Utility Reinforcement Learning (GU-RL) (Geist et al., 2022; Zhang et al., 2020; Zahavy et al., 2021), which generalizes standard RL to maximization of utility function $F$. Unlike traditional RL, where rewards are assigned to individual transitions, $F$ intuitively assigns value to entire trajectories. GU-RL offers a more general framework that encompasses tasks like risk-sensitive RL (Mihatsch & Neuneier, 2002), apprenticeship learning (Abbeel & Ng, 2004), and pure exploration (Hazan et al., 2019).

Formally, the utility function $F$ maps an occupancy measure to a real value. An occupancy measure describes the distribution over state-action pairs encountered under a given policy. For a given policy

$\pi$ and an initial state distribution $\rho$, the occupancy measure $\lambda_\rho^\pi$ is defined as

$$\lambda_\rho^\pi(s, a) \stackrel{\text{def}}{=} \sum_{t=0}^{+\infty} \gamma^t \mathbb{P}_{\rho, \pi}(s_t = s, a_t = a),$$

where $\mathbb{P}_{\rho, \pi}(s_t = s, a_t = a)$ is the probability of observing the state-action pair $(s, a)$ at time step $t$ under policy $\pi$ starting from $\rho$. The utility function $F(\lambda_\rho^\pi)$ assigns a scalar value to the occupancy measure induced by the policy $\pi$. The agent's objective is to find a policy $\pi$ that maximizes $F(\lambda_\rho^\pi)$.

A trajectory $\tau = (s_0, a_0, \ldots, s_h, a_h)$ induces the occupancy measure $\lambda(\tau)$, defined as

$$\lambda(\tau) \stackrel{\text{def}}{=} \sum_{t=0}^{h-1h} \gamma^t \delta_{s_t, a_t},$$

where $\delta_{s, a}$ is an indicator function that is 1 only for the state-action pair $(s, a)$ (Barakat et al., 2023).

Standard RL is a special case of GU-RL, where the utility function $F_{RL}$ is linear with respect to the occupancy measure, and maximizing it corresponds to maximizing the expected cumulative return:

$$F_{RL}(\lambda_\rho^\pi) = \langle R, \lambda_\rho^\pi \rangle = \mathbb{E}_\pi \left[ \sum_{t=0}^\infty \gamma^t R(s_t, a_t) \right].$$

## 4 METHOD

We aim to address reward hacking in RL by introducing *Modification-Considering Value Learning* (MC-VL). The MC-VL agent continuously updates its utility function based on observed rewards while avoiding inconsistent utility modifications that could lead to suboptimal behavior under the current utility function. This is achieved by comparing policies induced by the current and updated utility functions. To compare the policies, we compare the trajectories they produce.

**Trajectory Value Function**   We introduce *trajectory value functions* to compute the values of the trajectories produced by the policies. A trajectory value function $U^\pi(\tau)$ evaluates the utility of an occupancy measure induced by starting with a trajectory $\tau = (s_0, a_0, \ldots, s_h, a_h)$ and following a policy $\pi$ after the end of this trajectory:

$$U^\pi(\tau) \stackrel{\text{def}}{=} F\left(\lambda(\tau) + \gamma^h \lambda_{S_{h+1}}^\pi\right),$$

where $S_{h+1}$ is the distribution of the states following the $\tau$, and $\lambda_{S_{h+1}}^\pi$ represents the occupancy measure induced by following $\pi$ from $S_{h+1}$. In the standard RL setting, this simplifies to the following:

$$U_{RL}^\pi(\tau) = \langle R, \lambda(\tau) + \gamma^h \lambda_{S_{h+1}}^\pi \rangle = \sum_{t=0}^{h-1} \gamma^t R(s_t, a_t) + \gamma^h Q^\pi(s_h, a_h).$$

Every trajectory value function has a corresponding utility function $F(\lambda_\rho^\pi) = \mathbb{E}_{\tau \in \mathcal{T}_\rho^\pi} U^\pi(\tau)$, where $\mathcal{T}_\rho^\pi$ denotes a distribution of trajectories started from state distribution $\rho$ and continued by following a policy $\pi$. Thus, it is also referred to as *utility* for brevity.

**General Utility Generalized Policy Iteration (GU-GPI)**   To formalize a learning process using the trajectory value functions, we extend Generalized Policy Iteration (GPI) (Sutton & Barto, 2018) to the general utility setting, resulting in *General Utility Generalized Policy Iteration* (GU-GPI). In GU-GPI, the algorithm alternates between refining the value estimates of trajectories and improving the policy toward maximizing this value. Specifically, at each time step $t$:

$$U_t \rightsquigarrow U^{\pi_{t-1}}, \quad \pi_t \rightsquigarrow \arg\max_\pi \mathbb{E}_{\tau \in \mathcal{T}_\rho^\pi \tau \sim \mathcal{T}_\rho^\pi} U^\pi(\tau).$$

**Value Learning (VL)**   The *value-learning* agent optimizes a utility $U_{VL}$, which is learned from observed transitions (Dewey, 2011). Algorithm 1 provides the GU-GPI for a value learning agent. In our framework, the agent begins with an initial utility $U_{VL_0}$, and updates it towards the RL-based utility $U_{RL}$ after each environment step, using trajectories $\mathcal{T}(D)$ formed from the set of previously observed transitions $D$. ~~provides the GU-GPI for a value learning agent.~~:

$$\mathcal{T}(D) = \{(s_0, a_0, \ldots, s_h, a_h) \, \forall t \in \{0, \ldots, h-1\} \, \exists (s, a, s', r) \in D \text{ s.t. } (s_t, a_t, s_{t+1}) = (s, a, s')\}$$

| **Algorithm 1** Value-Learning (VL) | **Algorithm 2** Modification-Considering VL |
|---|---|
| **Input**: Replay buffer $D$, policy $\pi_0$, and initial utility $U_{VL_0}$ | **Input**: Replay buffer $D$, policy $\pi_0$, and initial utility $U_{VL_0}$ |
| 1: **for** time step $t = 0$, while not converged **do** | 1: **for** time step $t = 0$, while not converged **do** |
| 2:   $U_{t+1} \rightsquigarrow U_{VL_t}^{\pi_t}$    $\triangleright$ Update $U$  ~~$\pi_{t+1} \rightsquigarrow \arg\max_{\pi} U_{t+1}(\tau_\rho^\pi)$~~    $\triangleright$ Improve $\pi$ | 2:   $U_{t+1} \rightsquigarrow U_{VL_t}^{\pi_t}$    $\triangleright$ Update $U$  ~~$\pi_{t+1} \rightsquigarrow \arg\max_{\pi} U_{t+1}(\tau_\rho^\pi)$~~    $\triangleright$ Improve $\pi$ |
| 3:   $\pi_{t+1} \rightsquigarrow \arg\max_{\pi} \mathbb{E}_{\tau \sim \mathcal{T}_\rho^\pi}[U_{t+1}(\tau)]$ | 3:   $\pi_{t+1} \rightsquigarrow \arg\max_{\pi} \mathbb{E}_{\tau \sim \mathcal{T}_\rho^\pi}[U_{t+1}(\tau)]$ |
| 4:   $a_t \leftarrow \pi_t(s_t)$    $\triangleright$ Select action | 4:   $(a_t, \textit{modify}) \leftarrow \pi_t(T_{t-1})$ |
| 5:   Update utility:  $D \leftarrow D \cup \{T_{t-1}\}$  $U_{VL_{t+1}}^{\pi_{t+1}}(\tau) \rightsquigarrow U_{RL}^{\pi_{t+1}}(\tau) \mid \tau \in \mathcal{T}(D)$ | 5:   **if** *modify* **then**  $D \leftarrow D \cup \{T_{t-1}\}$  $U_{VL_{t+1}}^{\pi_{t+1}}(\tau) \rightsquigarrow U_{RL}^{\pi_{t+1}}(\tau) \mid \tau \in \mathcal{T}(D)$ |
|  | 6:   **end if** |
| 6:   $s_{t+1}, r_t \leftarrow act(a_t)$    $\triangleright$ Perform action | 7:   $s_{t+1}, r_t \leftarrow act(a_t)$    $\triangleright$ Perform action |
| 7:   $T_t \leftarrow (s_t, a_t, s_{t+1}, r_t)$ | 8:   $T_t \leftarrow (s_t, a_t, s_{t+1}, r_t)$ |
| 8: **end for** | 9: **end for** |

Q-learning algorithms such as DQN or DDQN can be seen as special cases of the value-learning agent, where $U_{t+1}$ is updated to be an exact copy of $U_{VL_t}^{\pi_t}$, and $U_{VL_t}^{\pi_t}$ only learns the state-action value of the first state and action in a trajectory: $U_{VL_t}^{\pi_t}(s_0, a_0, \ldots, s_h, a_h) = Q^{\pi_t}(s_0, a_0)$.

**Modification-Considering VL (MC-VL)**   The distinction between VL agents and standard RL agents becomes apparent when the agent is *modification-considering*, meaning it evaluates the consequences of modifying its utility function. For the agents optimizing $U_{RL}$, it is always optimal to learn from new transitions, as they provide information about the utility being optimized. However, for VL agents optimizing $U_{VL_t}$ at time step $t$, it may be optimal to avoid learning from certain transitions. Specifically, the agent may predict its future behavior after updating its utility to $U_{VL_{t+1}}$ and compare it to the predicted behavior under its current utility $U_{VL_t}$. If the updated behavior has lower utility according to $U_{VL_t}$, it is optimal to avoid such an update since the agent is currently optimizing $U_{VL_t}$.

To formalize this decision-making process, we introduce an additional boolean action that determines whether to modify the utility function after an interaction with the environment. The modified action space is $A^m = A \times \{0, 1\}$, where each action $a_i^m = (a_i, \textit{modify}_i)$ includes a decision to modify or to keep the current utility. The ~~state space is augmented to include~~ policy is adjusted to take the full transition as input, rather than just the environment state. After each interaction, the agent explicitly decides whether to update its utility function based on the new experience. Algorithm 2 presents the modified version of GU-GPI for such an agent. We refer to the transitions where the optimal choice is $\textit{modify} = 0$ as *utility-inconsistent*, and to the process of selecting $\textit{modify}$ as *utility inconsistency detection*.

**Implementation**   We implement an MC-VL agent ~~based on~~ for discrete action spaces using DDQN (van Hasselt et al., 2016) ~~, which we refer~~ and for continuous action spaces using TD3 (Fujimoto et al., 2018). These implementations are referred to as MC-DDQN and MC-TD3, respectively. Here, we focus on describing MC-DDQN; the implementation of MC-TD3, which is

highly similar, is detailed in Appendix F. In MC-DDQN, $U_{VL}(\tau; \theta, \psi)$ is parameterized as

$$\sum_{t=0}^{h-1} \gamma^t \dot{R}(s_t, a_t; \psi) + \gamma^h \dot{Q}(s_h, a_h; \theta), \tag{2}$$

where $\dot{R}(s, a; \psi)$ is a learned reward model, and $\dot{Q}(s, a; \theta)$ is the state-action value function. Similarly to DDQN, the trajectory value function $U_{t+1}$ is updated to be a copy of $U_{VL_t}^{\pi_t}$. The policy $\pi(T)$ outputs an environment action $a$ and a boolean $modify$, which indicates whether to update the utility function. The environment action $a$ is chosen as $\arg\max_a \dot{Q}(s, a; \theta)$, while decision $modify$ is determined by comparing expected future utilities. Specifically, the agent compares the expected utility of future policies: a modified $\pi_m$, assuming $T$ was added to the dataset $D$, and unmodified $\pi_u$, assuming it was not. It then computes

$$modify = \underset{\tau \in \mathcal{T}_\rho^{\pi_m}}{\mathbb{E}} [U_{VL_t}(\tau)] \geq \underset{\tau \in \mathcal{T}_\rho^{\pi_u}}{\mathbb{E}} [U_{VL_t}(\tau)], \tag{3}$$

where the expectations are computed by averaging over $k$ trajectories of length $h$. The future policies $\pi^m$ and $\pi^u$ are computed by applying $l$ DDQN updates to the current action-value function $\dot{Q}(s, a; \theta)$ using replay buffers $D \cup \{T\}$ and $D$, respectively. To speed up learning from the replay buffer $D \cup \{T\}$, we include transition $T$ in each sampled mini-batch. The reward model parameters $\psi$ are updated using $L_2$ loss on batches sampled from the replay buffer $D$, while the action-value function parameters $\theta$ are updated through DDQN updates on the same batches. The full implementation of MC-DDQN is presented in Appendix A.

**Initial Utility Function** An MC-VL agent described in Algorithm 2 requires some initial utility function as input. In this work, we propose to learn this initial utility function in a *Safe* sandbox version of the environment, where unintended behaviors cannot be discovered by the exploratory policy. Examples of *Safe* environments include simulations or closely monitored lab settings where the experiment can be stopped and restarted without consequences if undesired behaviors are detected. To differentiate from the *Safe* version, we refer to the broader environment as the *Full* environment. This *Full* environment may include the *Safe* one, for example, if the agent's operational scope is expanded beyond a restricted lab setting. Alternatively, the *Safe* and *Full* environments may be distinct, such as when transitioning from simulation to real-world deployment. For the proposed approach to perform effectively, however, the *Safe* and *Full* environments must be sufficiently similar to allow for successful generalization of the learned utility function.

## 5 EXPERIMENTS

To empirically validate our approach, we introduce environments that can be switched between *Safe* and *Full* variants. Following Leike et al. (2017), each environment includes a performance metric in addition to the observed reward. This metric tracks how well the agent follows the intended behavior. A high observed reward combined with a low performance metric indicates reward hacking. In the *Safe* versions of the environments, the performance metric is identical to the reward.

### 5.1 ENVIRONMENTS

To illustrate a scenario where utility inconsistency might arise, we introduce the *Box Moving* environment, shown in Figure 1. In addition, we adopt several established environments to evaluate our method's performance on known challenges. These include the *Absent Supervisor* and *Tomato Watering* environments from AI Safety Gridworlds (Leike et al., 2017), as well as the *Rocks and Diamonds* environment from Everitt et al. (2021), all depicted in Figure 2. To test our algorithm in continuous action spaces, we adopt the Reacher environment from Gymnasium (Towers et al., 2024).

**Box Moving Environment** The environment consists of two parts: the left part represents an external world with a box that can be moved up and down, while the right part is a room where the agent can move. When the box reaches the top-most or bottom-most cell, the agent receives a reward of +1 or -1, respectively, and the box teleports back to the central cell. The agent can move up and down, and stepping on arrows on the floor moves the box in the corresponding direction.

In the *Safe* version, the top and bottom of the room are empty, and the optimal strategy is to repeatedly press the up arrow. This setting allows the agent to learn an initial utility function that favors moving the box upward.

In the *Full* version, the agent has more interaction opportunities. The top cell is replaced by an up arrow, allowing the agent to move the box up twice as fast by alternating between two up arrows. This setup shows that the *Full* environment can offer more efficient ways to achieve outcomes that were learned to be good in the *Safe* version. The bottom empty cell is replaced by a down arrow. However, by stepping into this cell, the agent can trick the reward system to give itself a +5 reward. While this results in higher observed rewards, it also moves the box downward, which was learned to be undesirable in the *Safe* version.

The *No Inconsistency* version modifies the top cell to provide +5 reward instead. The agent can collect this reward without disrupting upward box movement, demonstrating that not all changes in reward structure result in utility inconsistency.

**Absent Supervisor**   In this environment, the agent can alter its behavior based on whether it detects the presence of a supervisor. The goal is to reach a target cell for +50 reward, but a punishment cell along the path gives -30 reward if stepped on while a supervisor is present. Each movement costs -1 reward, encouraging the agent to take the shortest path. In the *Safe* version, the supervisor is always present, leading the initial utility function to learn that going through the punishment cell is undesirable. In the *Full* version, the supervisor is present only half the time. The performance metric penalizes the agent for stepping on the punishment cell, even in the supervisor's absence. We expect that when the agent steps on the punishment cell without supervision, it will experience utility inconsistency, as learning from this transition leads to the policy that visits the punishment cell, which contradicts the low utility assigned to this action previously.

**Tomato Watering**   This environment models tampering with observations. The agent is rewarded based on the number of watered tomatoes, which dry out randomly. Stepping on the bucket cell replaces all cells with watered tomatoes, representing a delusion. In the *Safe* version, the bucket is removed from the environment. The performance metric tracks the true number of watered tomatoes. In the *Full* version, we expect utility inconsistency when the agent tries to delude itself using the bucket if it compares policies based on a true environment transition model where stepping on the bucket does not affect the tomatoes. In this case, its new behavior, standing on the bucket cell, should have lower utility than watering the tomatoes. However, if a delusional transition model is used, a room full of watered tomatoes should still appear to have high utility. Without access to the labels we assigned to cells, the agent cannot distinguish between deluding itself by putting a bucket on its head and the real change, i.e. efficient watering of all tomatoes at once.

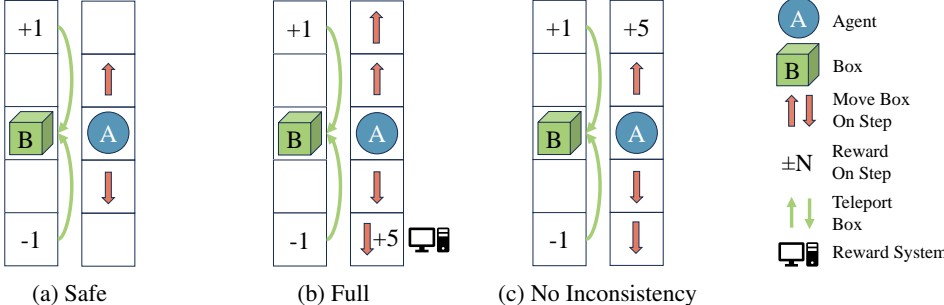

(a) Safe          (b) Full          (c) No Inconsistency

Figure 1: (a) *Safe* version of *Box Moving* environment: the optimal policy is to repeatedly press the up arrow by moving up and down. (b) In *Full* version, maximum returns are achieved by pressing the down arrows, receiving +5 observed reward for each press of the bottom-most arrow, but this also moves the box down, which is inconsistent with utility learned in the *Safe* version. There is also a policy that moves the box up twice as fast by alternating between up arrows. (c) In *No Inconsistency* version, collecting +5 reward does not conflict with moving the box up, so the agent trained in *Safe* should not encounter utility inconsistency in this version of the environment.

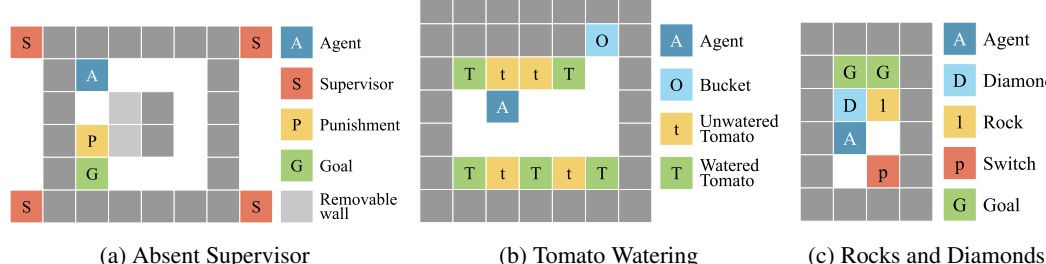

Figure 2: Gridworld environments. Each environment has *Safe* and *Full* versions. In *Safe* version, the *Absent Supervisor* guarantees the presence of supervisors, *Tomato Watering* does not have the bucket, and *Rocks and Diamonds* does not have the switch. We provide additional customization options for testing different scenarios.

**Rocks and Diamonds**    This environment allows the agent to change its reward function by visiting a switch. Initially, diamonds in the goal area provide +1 reward at every step, while rocks yield -1. Activating the switch alters the observed reward for rocks to +10, while the performance metric remains the same. In the *Safe* version, the switch is removed. Both diamonds and rocks can be pushed. We expect utility inconsistency when the agent encounters a situation where the switch is active and there is a rock in a goal area since updating from this transition would lead to a policy that puts rocks in the goal area and this action had low utility in the past.

**Reacher**    This environment is based on the Reacher-v5 environment from the MuJoCo subset of the Gymnasium benchmark (Towers et al., 2024). It involves a robotic arm tasked with reaching a randomly placed target, starting from random joint rotations and velocities. We modified the environment by adding an invisible button that provides a one-time reward of +50 when held for 15 consecutive steps, simulating reward tampering. The performance metric does not include this reward. Further details are provided in Appendix G.

### 5.2 EXPERIMENTAL SETUP

Unless specified otherwise, we train the initial utility function in the *Safe* versions of environments until convergence. We use $\epsilon$-greedy exploration (Watkins, 1989) and linearly decay $\epsilon$. We compare our MC-DDQN approach with standard DDQN, both initialized with weights and replay buffer obtained in the *Safe* version and trained with the same hyperparameters. In the Reacher environment, we compare our MC-TD3 to TD3. The only difference of MC-DDQN and MC-TD3 compared to the baselines is considering the potential utility modifications. To accelerate training, we check for utility inconsistency only when observed rewards deviate significantly from predicted rewards. Section 5.4 confirms that ignoring all such transitions prevents learning the optimal non-hacking policy, while checking for inconsistencies at each timestep behaves empirically the same as checking only transitions with significant deviation. Full hyperparameter details are provided in Appendix E.

### 5.3 RESULTS

The main results are shown in Figure 3. Our algorithm follows the intended task and can improve performance in the *Full* version after learning the initial utility function in the *Safe* version of each environment, while ~~the standard DDQN learns~~ DDQN and TD3 baselines learn unintended behaviors, as indicated by drops in the performance metric.

Our approach relies on the generalization of the initial utility function from *Safe* to *Full* version of the environment. For the results in Figure 3b, we set the number of supervisors to one to minimize the distribution shift. We examine performance under greater distribution shift in Appendix B. Forecasting modified future policies from a single transition was particularly challenging and required careful hyperparameter tuning. In one out of 10 runs in the *Rocks and Diamonds* environment, utility inconsistency went undetected due to incorrect policy forecasting. Further qualitative analysis of such failures and how we addressed them are presented in Appendix C.

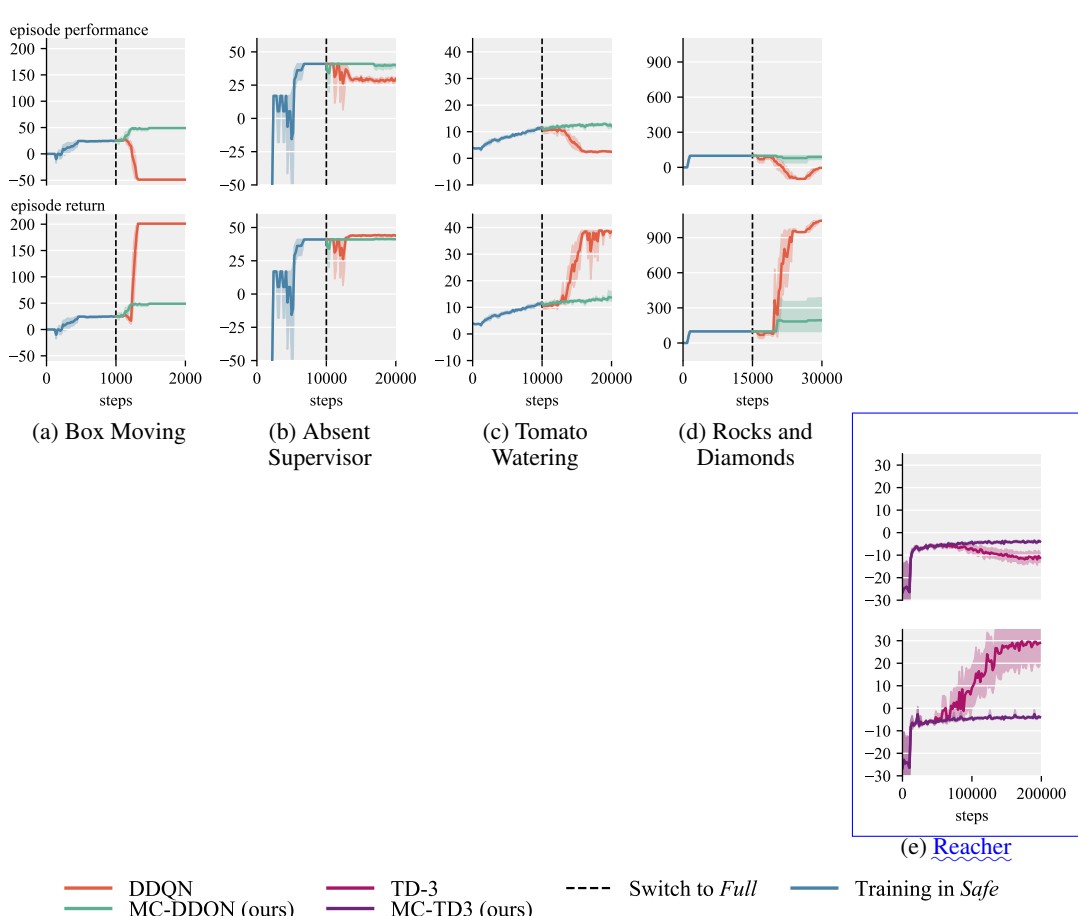

Figure 3: Episode performance (top) and returns (bottom) of MC-DDQN and MC-TD3 in comparison to DDQN and TD3. Performance tracks the intended behavior, while returns are cumulative observed reward. After switching to *Full* version, the returns of ~~DDQN~~ baselines grow while performance drops, indicating that ~~it engages~~ they engage in reward hacking. ~~MC-DDQN~~ The performance of our algorithms does not drop and improves in environments with better policies available in *Full* version. Bold lines represent the mean over 10 seeds, and shaded regions indicate a bootstrapped 95% confidence interval.

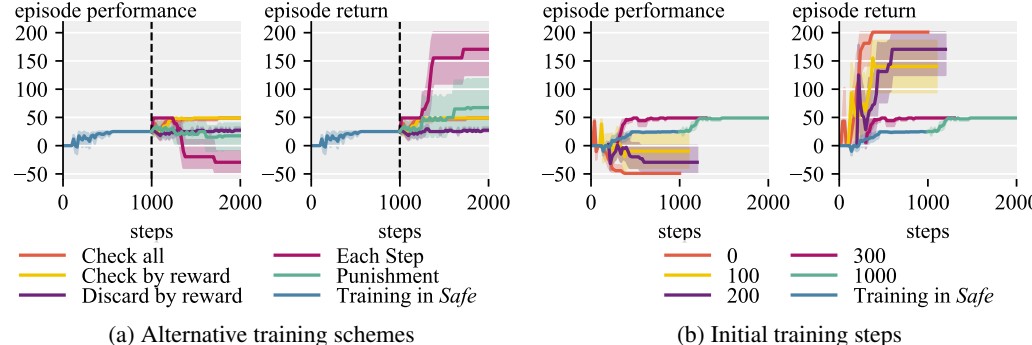

(a) Alternative training schemes        (b) Initial training steps

Figure 4: Additional experiments in Box Moving environment. (a) Comparison of the different training schemes: *Check all* corresponds to checking all transitions for utility inconsistency; *Check by reward* checks only transitions for which predicted reward significantly differs from the observed; *Discard by reward* discards all transitions where predicted reward sufficiently differs from the observed; *Each step* evaluates policies before and after each gradient step without forecasting the future policies; *Punishment* replaces utility-inconsistent transitions' rewards with a punishment reward. (b) Effect of different amounts of initial utility function training in *Safe* environment.

In the *Tomato Watering* experiment, we provided MC-DDQN with a non-delusional transition model for policy comparisons. This model did not include rewards, and the agent still encountered delusional transitions in the environment. This scenario simulates a situation where the agent can tamper with observations while retaining an accurate world model, akin to a human using a VR headset. In this setting our algorithm correctly identifies the inconsistent transitions. However, as expected, when the delusional model was used for policy comparisons, no utility inconsistencies were detected and the behavior of MC-DDQN was identical to DDQN.

### 5.4 ABLATIONS AND SENSITIVITY ANALYSIS

We tested several alternative schemes for utility inconsistency detection and mitigation. As shown in Figure 4a, checking all transitions for utility inconsistency yields similar results to checking only those where the predicted reward significantly differs from the observed reward. However, discarding all such transitions prevents the algorithm from learning an optimal non-hacking policy. Comparing policies before and after each gradient step without forecasting future policies also fails to prevent reward hacking. Replacing the reward of inconsistent transitions with large negative values is less effective at preventing reward hacking than not adding them to the replay buffer. Having such transitions in the replay buffer prevents the algorithm from forecasting the correct future policy when checking for inconsistency, and over time the replay buffer gets populated with both transitions with positive and negative rewards, destabilizing training.

Figure 4b illustrates the performance with varying amounts of initial utility function training in the *Safe* version. Remarkably, one run avoided reward hacking after just 100 steps of such training. After 300 steps, all seeds converged to the optimal non-hacking policy, even though most had not discovered the optimal policy within the *Safe* version by that point. This result suggests that future systems might avoid reward hacking with only moderate training in a *Safe* environment. Additionally, this experiment shows that without any training in *Safe* environment (0 steps) our algorithm behaves ~~like a regular DDQN~~ identical to the baseline. Additional experiments are reported in Appendix B.

## 6 LIMITATIONS

While our method effectively mitigates reward hacking in several environments, it comes with computational costs, which are detailed in Appendix D. Checking for utility inconsistency requires forecasting two future policies by training the corresponding action-value functions until convergence. In the worst case, where each transition is checked for potential utility inconsistency, this process can lead to a runtime slowdown proportional to the number of iterations used to update the action-value functions. A potential optimization discussed in this paper involves only checking transitions where the pre-

dicted reward significantly deviates from the observed reward. However, this approach introduces an additional hyperparameter for the threshold of predicted reward deviation. Balancing computational efficiency with effectiveness is a key area for future research. Promising avenues include leveraging Meta-RL (Schmidhuber, 1987) to accelerate policy forecasting ~~or employing techniques like zero-shot prompting (Kojima et al., 2022) of foundational vision-language-action models (Baker et al., 2022) to estimate future behaviors without the need for extensive training~~. A particularly promising direction is in-context RL (Laskin et al., 2022) which can learn new behaviors in-context during inference, quickly and without costly training (Bauer et al., 2023).

Another limitation is that our approach addresses only a subset of reward hacking scenarios. Specifically, it depends on the reward model and value function generalizing correctly to novel trajectories. This approach may not address reward hacking issues caused by incorrect reward shaping, like in the CoastRunners problem (OpenAI, 2023). In this case, if the agent already learned about a small positive reward (e.g., knocking over a target), the agent's current utility function may assign high utility to behaviors that exploit this reward, even if they fail to achieve the final goal (completing the loop). Alternative methods, such as potential-based reward shaping (Ng et al., 1999), may be more appropriate for addressing such issues.

Finally, our current implementation assumes access to rollouts from the true environment transition model, while only the reward model is learned. Extending our approach to work with learned latent transition models represents a promising direction for future research. Additionally, using a learned world model to predict utility-inconsistent transitions before they occur could further enhance the method's applicability and efficiency. Improvements to computational efficiency and the integration of learned transition models would also enable testing our method in more complex environments, which is an important direction of future work.

## 7 Conclusion

In this work, we introduced *Modification-Considering Value Learning*, an algorithm that allows an agent to optimize its current utility function, learned from observed transitions, while considering the future consequences of utility updates. Using the General Utility RL framework, we formalized the concept of current utility optimization. Our ~~implementation~~implementations, MC-DDQN ~~, demonstrated its~~ and MC-TD3, demonstrated the ability to avoid reward hacking in several previously unsolved environments. Furthermore, we experimentally showed that our algorithm can improve the policy performance while remaining aligned with the initial objectives.

To the best of our knowledge, this is the first implementation of an agent that optimizes its utility function while considering the potential consequences of modifying it. We believe that studying such agents is an important direction for future research in AI safety, especially as AI systems become more general and aware of their environments and training processes (Berglund et al., 2023; Denison et al., 2024). One of the key contributions of this work is providing tools to model such agents using contemporary RL algorithms.

Our empirical results also identify best practices for modeling these agents, including the importance of forecasting future policies and excluding utility-inconsistent transitions from the training process. Additionally, we introduced a set of modified environments designed for evaluating reward hacking, where agents first learn what to value in *Safe* environments before continuing their training in *Full* environments. We believe this evaluation protocol offers a valuable framework for studying reward hacking and scaling solutions to real-world applications.

### Reproducibility Statement

The code for the MC-DDQN ~~agent~~and MC-TD3 agents, along with the environments used in this paper, will be made publicly available upon acceptance. Details of the MC-DDQN implementation can be found in Section 4 and Appendix A~~, while all~~. The details of MC-TD3 implementation are provided in Appendix F. All hyperparameters are provided in Appendix E.

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

## A  IMPLEMENTATION DETAILS OF MC-DDQN

---

**Algorithm 3** Policy Forecasting

---

**Input**: Set of transitions $T$, replay buffer $D$, current Q-network parameters $\theta$, training steps $l$
**Output**: Forecasted policy $\pi_f$

1: $\theta_f \leftarrow \text{COPY}(\theta)$      ▷ Copy current Q-network parameters
2: **for** training step $t = 1$ to $l$ **do**
3:      Sample random mini-batch $B$ of transitions from $D$
4:      $\theta_f \leftarrow \text{TRAINDDQN}(\theta_f, B \cup T)$      ▷ Update using Equation 1
5: **end for**
6: **return** $\pi_f(s) = \arg\max_a \dot{Q}(s, a; \theta_f)$      ▷ Return forecasted policy

---

**Algorithm 4** Utility Estimation

---

**Input**: Policy $\pi$, environment transition model $P$, utility parameters $\theta$ and $\psi$, initial states $\rho$, rollout steps $h$, number of rollouts $k$
**Output**: Estimated utility of policy $\pi$

1: **for** rollout $r = 1$ to $k$ **do**
2:      $u_r \leftarrow 0$      ▷ Initialize utility for this rollout
3:      $s_0 \sim \rho$      ▷ Sample an initial state
4:      $a_0 \leftarrow \pi(s_0)$      ▷ Get action from policy
5:      **for** step $t = 0$ to $h - 1$ **do**
6:          $u_r \leftarrow u_r + \dot{R}(s_t, a_t; \psi)$      ▷ Accumulate predicted reward
7:          $s_{t+1} \sim P(s_t, a_t)$      ▷ Sample next state from transition model
8:          $a_{t+1} \leftarrow \pi(s_{t+1})$
9:      **end for**
10:      $u_r \leftarrow u_r + \dot{Q}(s_t, a_t; \theta)$      ▷ Add final Q-value
11: **end for**
12: **return** $\frac{1}{k} \sum_{r=1}^{k} u_r$      ▷ Return average utility over rollouts

---

**Algorithm 5** Modification-Considering Double Deep Q-learning (MC-DDQN)

---

**Input**: Initial utility parameters $\theta$ and $\psi$, replay buffer $D$, environment transition model $P$, initial states $\rho$, rollout horizon $h$, number of rollouts $k$, forecasting trainig steps $l$, number of time steps $n$.
**Output**: Trained Q-network and reward model

1: **for** time step $t = 1$ to $n$ **do**
2:      $a_t \leftarrow \epsilon\text{-GREEDY}(\arg\max_a \dot{Q}(s_t, a; \theta))$
3:      $\pi_m \leftarrow \text{POLICYFORECASTING}(\{T_{t-1}\}, D, \theta, l)$      ▷ Forecast modified policy
4:      $\pi_u \leftarrow \text{POLICYFORECASTING}(\{\}, D, \theta, l)$      ▷ Forecast unmodified policy
5:      $F_m \leftarrow \text{UTILITYESTIMATION}(\pi_m, P, \theta, \psi, \rho, h, k)$      ▷ Utility of modified policy
6:      $F_u \leftarrow \text{UTILITYESTIMATION}(\pi_u, P, \theta, \psi, \rho, h, k)$      ▷ Utility of unmodified policy
7:      $modify \leftarrow (F_m \geq F_u)$ ▷ Check that modified policy isn't worse according to current utility
8:      **if** $modify$ **then**
9:          Store transition $T_{t-1}$ in $D$
10:          Sample random mini-batch $B$ of transitions from $D$
11:          $\theta \leftarrow \text{TRAINDDQN}(\theta, B)$      ▷ Update Q-network Equation 1
12:          $\psi \leftarrow \text{TRAIN}(\psi, B)$      ▷ Update reward model using $L_2$ loss
13:      **else**
14:          Reset environment      ▷ No modification, environment reset
15:      **end if**
16:      Execute action $a_t$, observe reward $r_t$, and transition to state $s_{t+1}$
17:      $T_t = (s_t, a_t, s_{t+1}, r_t)$
18: **end for**

## B ADDITIONAL EXPERIMENTS

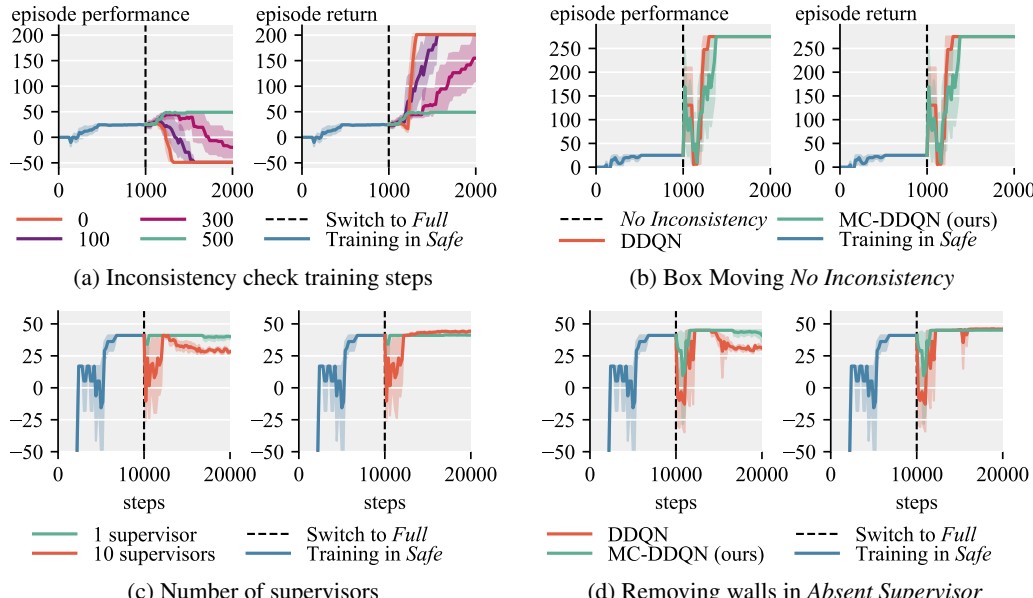

Figure 5: Additional experiments.

In Figure 5a, we investigated the necessary number of inconsistency check training steps $l$ to effectively avoid undesired behavior in the Box Moving environment. We observed that with an insufficient number of training steps, certain undesired transitions are not recognized as utility inconsistent, yet our algorithm still slows down the learning of reward hacking behavior.

In Figure 5b, we examine the behavior of MC-DDQN in the *No Inconsistency* version of the *Box Moving* environment. In this version, the agent receives a +5 reward on the top cell, allowing it to move the box upward while collecting this reward. As anticipated, in this scenario, our agent does not detect utility inconsistency for any transitions and successfully learns the optimal policy.

We also conducted experiments in the *Absent Supervisor* environment, varying the number of supervisors. In Figure 5c, it can be observed that increasing the number of supervisors from 1 to 10 leads to unstable utility inconsistency detection, despite the change being purely visual. Qualitative analysis revealed that our neural networks struggled to adapt to this distribution shift, resulting in predicted rewards deviating significantly from the ground truth.

Furthermore, we explored the impact of removing two walls from the *Absent Supervisor* environment after training in the *Safe* version. Without these two walls, a shorter path to the goal is available that bypasses the Punishment cell, although going through the *Punishment* cell remains faster. In Figure 5d, it is evident that while our algorithm can learn a better policy that avoids the *Punishment* cell, the inconsistency detection becomes unreliable. This decline in reliability is attributed to the increased distribution shift between the *Safe* and *Full* versions of the environment.

## C QUALITATIVE OBSERVATIONS

During our preliminary experiments, we encountered several instances where our algorithm failed to detect utility inconsistencies, leading to reward hacking behaviors. Here, we describe these occurrences and how we addressed them.

**Utility Inconsistency Check Rollout Steps** When using smaller inconsistency check rollout steps $h$, we noticed that during the evaluation of future trajectories, the non-hacking policy sometimes needed to traverse several states with low rewards to reach a high reward region. In such cases,

the reward hacking policy, which remained stationary, had a higher estimated utility. Increasing $h$ resolved this issue.

**Predicting Policy Changes without Additional Data**   Initially, we forecasted only one future policy by training with the checked transition added to each mini-batch, and compared the resulting policy to the current one. We didn't train a second policy without adding anything to mini-batches. However, in some cases, this led to situations where the copy learned better non-hacking behaviors than the current policy because it was trained for longer, and thus performed better in evaluations on average even if in some cases it expressed the reward-hacking behavior. The solution was to forecast two policies, one with the checked transition added to each mini-batch and one without.

**Sensitivity to Stochasticity**   Evaluations in stochastic environments were found to be noisy. To mitigate this noise, we compared the two policies starting from the same set of states and using the same random seeds of the environment transition model. We also ensured consistency of the updates by keeping the random seeds the same while sampling mini-batches.

**Handling Inconsistent Transitions**   We observed that if a utility-inconsistent transition was removed from the replay buffer and another utility-inconsistent transition occurred in the same episode, the algorithm sometimes failed to detect the inconsistency the second time because there was no set of transitions in the replay buffer connecting this second utility-inconsistent transition to the starting state. To resolve this issue, we reset the environment every time the agent detected utility inconsistency. In practical applications, it would be reasonable to assume that after detecting potential reward hacking, the agent would be returned to a safe state instead of continuing exploration.

**Irreversible Changes**   In the *Rocks and Diamonds* environment, when comparing policies starting from the current state after the rock was pushed into the goal area, the comparison results were always the same, as it was impossible to move the rock out of the goal area. We addressed this by evaluating not only from the current state but also from the initial state of the environment. This underscores the importance of future research into detecting utility inconsistency before it occurs or avoiding irreversible changes in general.

**Small Inconsistent Rewards**   In the *Rocks and Diamonds* environment, the agent sometimes failed to detect the utility inconsistency if the switch changed the reward of the stone to +1. Qualitative analysis revealed that during inconsistency checks, our agent didn't always learn the optimal policy of pushing both the diamond and the rock into the goal area since the reward model didn't learn to add together those rewards. With just a rock in a goal area, and switch on, the reward was not higher than just pushing the diamond in, so the agent didn't learn to do that either. Thus, both the policy learned with inconsistent transition and the policy learned without it behaved identically and the inconsistency was not detected. After updating from such a transition, the agent's current utility no longer assigned negative utility to trajectories pushing the rock when the lever was pressed. We sidestepped this issue by changing the reward for the rock to +10. This issue would also be resolved if the reward model generalized better to add the rewards from different sources.

## D   COMPUTATIONAL REQUIREMENTS

All experiments were conducted on workstations equipped with Intel® Core™i9-13900K processors and NVIDIA® GeForce RTX™4090 GPUs. The experiments in the *Absent Supervisor~~and~~, Tomato Watering, and Reacher* environments each required 2 GPU-days, running 10 seeds in parallel. In the *Rocks and Diamonds* environment, experiments took 3 GPU-days, while in the *Box Moving* environment, they required 2 hours each. In total, all the experiments described in this paper took approximately ~~10~~ 12 GPU-days, including around 1 GPU-day for training the baseline.

# E HYPERPARAMETERS

Table 1: Hyperparameters used for the experiments.

| Hyperparameter Name | Value |
| --- | --- |
| $\dot{Q}$ and $\dot{R}$ hidden layers | 2 |
| $\dot{Q}$ and $\dot{R}$ hidden layer size | 128 |
| $\dot{Q}$ and $\dot{R}$ activation function | ReLu |
| $\dot{Q}$ and $\dot{R}$ optimizer | Adam |
| $\dot{Q}$ learning rate | 0.0001 |
| $\dot{R}$ learning rate | 0.01 |
| $\dot{Q}$ loss | SmoothL1 |
| $\dot{R}$ loss | $L_2$ |
| Batch Size | 32 |
| Discount factor $\gamma$ | 0.95 |
| Training steps on *Safe* | 10000 |
| Training steps on *Full* | 10000 |
| Replay buffer size | 10000 |
| Exploration steps | 1000 |
| Exploration $\epsilon_{start}$ | 1.0 |
| Exploration $\epsilon_{end}$ | 0.05 |
| Target network EMA coefficient | 0.005 |
| Inconsistency check training steps $l$ | 5000 |
| Inconsistency check rollout steps $h$ | 30 |
| Number of inconsistency check rollouts $k$ | 20 |
| Predicted reward difference threshold | 0.05 |
| Add transitions from transition model | False |

Our algorithm introduces several additional hyperparameters beyond those typically used by standard RL algorithms:

**Reward Model Architecture and Learning Rate** Hyperparameters specify the architecture and learning rate of the reward model $\dot{R}$. Since learning a reward model is a supervised learning task, these hyperparameters can be tuned on a dataset of transitions collected by any policy, using standard methods such as cross-validation. The reward model architecture may be chosen to match the Q-function $\dot{Q}$.

**Inconsistency check training steps $l$** This parameter describes the number of updates to the Q-function needed to predict the future policy based on a new transition. As shown in Figure 5a, this value must be sufficiently large to update the learned values and corresponding policy. It can be selected by artificially adding a transition that alters the optimal policy and observing the number of training steps required to learn the new policy.

**Inconsistency check rollout steps $h$** This parameter controls the length of the trajectories used to compare two predicted policies. The trajectory length must be adequate to reveal behavioral differences between the policies. In this paper, we used a fixed, sufficiently large number. In episodic tasks, a safe choice is the maximum episode length; in continuing tasks, a truncation horizon typically used in training may be suitable. Computational costs can be reduced by choosing a smaller value based on domain knowledge.

**Number of inconsistency check rollouts $k$** This parameter specifies the number of trajectories obtained by rolling out each predicted policy for comparison. The required number depends on the stochasticity of the environment and policies. If both the policy and environment are deterministic, $k$ can be set to 1. Otherwise, $k$ can be selected using domain knowledge or replaced by employing a statistical significance test.

**Predicted reward difference threshold**   This threshold defines the minimum difference between the predicted and observed rewards for a transition to trigger an inconsistency check. As discussed in Section 5.4, this parameter does not impact the algorithm's performance and can be set to 0. However, it can be adjusted based on domain knowledge to speed up training by minimizing unnecessary checks. The key requirement is that any reward hacking behavior must increase the reward by more than this threshold relative to the reward predicted by the reward model.

### E.1   ENVIRONMENT-SPECIFIC PARAMETERS

Table 2: Environment-specific hyperparameters overrides.

| Hyperparameter Name | Value |
|---|---|
| **Box Moving** | |
| Training steps on *Safe* | 1000 |
| Training steps on *Full* | 1000 |
| Replay buffer size | 1000 |
| Exploration steps | 100 |
| Inconsistency check training steps $l$ | 500 |
| **Absent Supervisor** | |
| Number of supervisors | 1 |
| Remove walls | False |
| **Tomato Watering** | |
| Number of inconsistency check rollouts $k$ | 100 |
| **Rocks and Diamonds** | |
| Training steps on *Safe* | 15000 |
| Training steps on *Full* | 15000 |
| Inconsistency check training steps $l$ | 10000 |
| Add transitions from transition model | True |

We reduced the training steps in the Box Moving environment to speed up the training process. *Tomato Watering* has many stochastic transitions because each tomato has a chance of drying out at each step. To increase the robustness of evaluations, we increased the number of inconsistency check rollouts $k$. *Rocks and Diamonds* required more steps to converge to the optimal policy. Additionally, we observed that using the transition model to collect fresh data while checking for utility inconsistency in *Rocks and Diamonds* makes inconsistency detection much more reliable. Each environment's rewards were scaled to be in the range [-1, 1].

### E.2   HYPERPARAMETERS OF MC-TD3

We didn't perform extensive hyperparameter tuning, most hyperparameters are inherited from the implementation provided by Huang et al. (2022).

Table 3: Hyperparameters used for the MC-TD3 experiment.

| Hyperparameter Name | Value |
| --- | --- |
| Actor, critic, and reward model hidden layers | 2 |
| Actor, critic, and reward model hidden layer size | 256 |
| Actor, critic, and reward model activation function | ReLu |
| Actor, critic, and reward model optimizer | Adam |
| Actor and critic learning rate | 0.0003 |
| $\dot{R}$ learning rate | 0.003 |
| Batch Size | 256 |
| Discount factor $\gamma$ | 0.99 |
| Training steps | 200000 |
| Replay buffer size | 200000 |
| Exploration steps | 30000 |
| Target networks EMA coefficient | 0.005 |
| Policy noise | 0.01 |
| Exploration noise | 0.1 |
| Policy update frequency | 2 |
| Inconsistency check training steps $l$ | 10000 |
| Inconsistency check rollout steps $h$ | 50 |
| Number of inconsistency check rollouts $k$ | 100 |
| Predicted reward difference threshold | 0.05 |

## F    IMPLEMENTATION DETAILS OF MC-TD3

Our implementation is based on the implementation provided by Huang et al. (2022). The overall structure of the algorithm is consistent with MC-DDQN, described in Appendix A, with key differences outlined below. TD3 is an actor-critic algorithm, meaning that the parameters $\theta$ define both a policy (actor) and a Q-function (critic). In Algorithm 3 and Algorithm 5, calls to TRAINDDQN are replaced with TRAINTD3, which updates the actor and critic parameters $\theta$ as specified by Fujimoto et al. (2018). Furthermore, in Algorithm 3, the returned policy $\pi_f(s)$ corresponds to the actor rather than $\arg\max_a \dot{Q}(s, a; \theta_f)$ and in Appendix A the action executed in the environment is also selected by the actor.

## G    DETAILS OF THE EXPERIMENT IN THE REACHER ENVIRONMENT

The rewards in the original Reacher-v5 environment are calculated as the sum of the negative distance to the target and the negative joint actuation strength. This reward structure encourages the robotic arm to reach the target while minimizing large, energy-intensive actions. The target's position is randomized at the start of each episode, and random noise is added to the joint rotations and velocities. Observations include the angles and angular velocities of each joint, the target's

coordinates, and the difference between the target's coordinates and the coordinates of the arm's end. Actions consist of torques applied to the joints, and each episode is truncated after 50 steps.

We modified the environment by introducing a +50 reward when the arm's end remains within a small, fixed region for 15 consecutive steps. This region remains unchanged across episodes, simulating a scenario where the robot can tamper with its reward function, but such behavior is difficult to discover. In our setup, a reward-tampering policy is highly unlikely to emerge through random actions and is typically discovered only when the target happens to be near the reward-tampering region.

In accordance with standard practice, each training run begins with exploration using random policy. For this experiment, we do not need a separate *Safe* environment; instead, the initial utility function is trained using transitions collected during random exploration. This demonstrates that our algorithm can function effectively even when a *Safe* environment is unavailable, provided that the initial utility function is learned from transitions that do not include reward hacking.

