# OpenReview forum: "Modification-Considering Value Learning for Reward Hacking Mitigation in RL"
_ICLR.cc/2025/Conference — Submitted to ICLR 2025_

### Official Review · Reviewer_oLm1 · 2024-10-19

**Soundness:** 4
**Presentation:** 3
**Contribution:** 3
**Rating:** 6
**Confidence:** 2

**Summary:**

The paper introduces Modification-Considering Value Learning (MC-VL) to address reward hacking in reinforcement learning, where agents exploit unintended strategies to maximize rewards without achieving the desired outcomes.
Using the General Utility RL (GU-RL) framework, MC-VL refines the agent's utility function by evaluating policy updates across entire trajectories, helping to avoid short-term reward maximization that harms long-term objectives.
Implemented with Double Deep Q-Networks (DDQN), the method prevents reward hacking in safety-critical environments, offering a novel solution for achieving long-term alignment in RL.

**Strengths:**

- The paper is well written.
- Reward hacking is one of the main reasons why reinforcement learning is not widely deployed in real-world applications; any contribution that tackles this issue is important for the community.
- Using the utility function to avoid short-term reward maximization that harms long-term objectives is an elegant way to avoid value hacking.

**Weaknesses:**

- The evaluations are solely on toy environments; it would be interesting to test this approach on more complex dynamics and reward functions, such as in the continuous control suite MuJoCo, where reward hacking and undesired behaviors could be more prevalent.

**Questions:**

- Could MC-VL be considered not only as an approach for avoiding reward hacking, but also as a potential method for improving exploration?

---

> ### Author Response · Authors · 2024-11-17
> **Official Reply to Reviewer oLm1**
>
> Thank you for your thoughtful review and valuable feedback!
>
> > it would be interesting to test this approach on more complex dynamics and reward functions, such as in the continuous control suite MuJoCo
>
> In response to this suggestion, we designed an additional experiment using the Reacher task from the MuJoCo suite, which is outlined in our common response. Please let us know if this additional experimental setup sufficiently addresses your concern!
>
> > Could MC-VL be considered not only as an approach for avoiding reward hacking, but also as a potential method for improving exploration?
>
> Thank you for this insightful question! We agree that reward hacking is often closely tied to exploration since it can emerge when an agent discovers unintended ways to achieve high rewards. Consequently, concepts from our paper may indeed lend themselves to enhancing exploration. For example, the discrepancy between predicted and observed rewards, as well as assessing how much a given transition impacts future behavior, could serve as useful signals for exploration. However, a full investigation of these ideas would likely exceed the scope of this work.
>
> Thank you again for your valuable review! Please let us know if our responses address your questions or if you have additional suggestions.

---

> > ### Comment · Reviewer_oLm1 · 2024-11-25
> >
> > I thank the authors for answering all my questions. I will keep my rating

---

> > > ### Author Response · Authors · 2024-11-28
> > > **Official Reply to Reviewer oLm1**
> > >
> > > Thank you for your response and for supporting the acceptance of our work. We were wondering what further changes you would like to see to be able to recommend acceptance and increase your score further.
> > >
> > > Thank you again for your efforts in reviewing our submission!

---

### Official Review · Reviewer_bLnS · 2024-10-23

**Soundness:** 2
**Presentation:** 2
**Contribution:** 3
**Rating:** 5
**Confidence:** 4

**Summary:**

This paper designs a novel algorithm Modification-Considering Value Learning (MC-VL) to address a phenomenon known as reward hacking in RL. MC-VL allows an agent to optimize its current utility function while considering the future consequences of utility updates, where the concept of current utility optimization is formalized using the General Utility RL framework. To the best of the authors’ knowledge, MC-VL is the first implementation of an agent that optimizes its utility function while considering the potential consequences of modifying it. The authors demonstrate the effectiveness of MC-VL in preventing reward hacking across various grid-world tasks, including benchmarks from the AI Safety Gridworlds suite.

**Strengths:**

1. The motivation is well justified, the method is reasonable
2. The algorithm addressed reward hacking through the lens of General Utility RL is novel and interesting
3. The empirical results show the effectiveness of the proposed algorithm, including benchmarks from the AI Safety Gridworlds suite.

**Weaknesses:**

1. MC-VL is based on the General Utility RL framework and the basic RL method for implementing MC-VL is only DDQN. Why did authors not try to implement MC-VL based on other RL methods, such as PPO or DDPG?
2. As stated in L65-L72, the robot grasping task is a typical case, why did authors not choose this environment to evaluate the performance? In addition, how does the proposed algorithm scale with a larger number of agents and more complex environments?
3. The comparison baseline method is only the basic RL method DDQN, it is better to compare more methods addressing reward hacking, such as the methods authors introduce in Related Work

**Questions:**

I have asked my questions in the Weaknesses section. I have no further questions.

---

> ### Author Response · Authors · 2024-11-17
> **Official Reply to Reviewer bLnS**
>
> Thank you for your thoughtful review and valuable feedback! In response to each of your points, we have provided detailed replies and designed an additional experiment to strengthen the empirical evaluation and address your concerns.
>
> > Why did authors not try to implement MC-VL based on other RL methods, such as PPO or DDPG
>
> To address this point, as well as similar concerns from other reviewers, we have conducted an additional experiment implementing MC-VL with TD3, an extension of DDQN. We chose TD3 because it is a good balance between performance and ease of implementation. Please let us know if this experiment, described in the common response, adequately addresses your concerns about our algorithm's empirical evaluation.
>
> > Robot grasping task is a typical case, why did authors not choose this environment to evaluate the performance?
>
> Instances of reward hacking in existing robotics literature do usually not focus on the reward hacking aspect specifically.  There, reward hacking behavior is typically regarded as an error and mitigated by adjusting the reward function to better reflect designer intentions. As a result, details and code for such experiments are rarely available, making these setups challenging to reproduce. In our additional experiment, we aim to address this concern by using an environment based on MuJoCo Reacher-v5, which similarly features a robotic arm. We would be happy to know whether this aligns with what the reviewer had in mind.
>
> > Compare more methods addressing reward hacking, such as the methods authors introduce in Related Work
>
> Most prior work on reward hacking is theoretical and lacks empirical evaluations. The work performing empirical evaluations is unfortunately considered settings not applicable to ours. In particular,
> - The work by Everitt and Hutter [1] requires a prior over manually specified utility functions and was only tested in a *tabular* setting. This is incompatible with our approach, as we rely on reward model generalization across similar states.
> - Einstein et al. [2] apply reward hacking techniques to *large language models* using a reward model learned from human feedback, which is a very different context from ours.
> - Laidlaw et al. [3] constrain the learned policy’s state-action occupancy measure to remain *close* to a known safe policy’s occupancy measure. This is a very conservative approach since it prevents the learned policy from visiting the states not visited by the safe policy. In such states, the KL-divergence term of their objective becomes infinite, preventing the policy from learning to access those states. In our experiments in Box Moving environment (Figure 3a) and Absent Supervisor environment without walls (Figure 5d), we show that our method is capable of learning a policy that visits new states, not visited in by the policy from the *Safe* environment. We do not provide a direct comparison to [3] because their implementation, to the best of our knowledge, is not available.
>
> Thank you again for your valuable feedback. Please let us know if our responses address your concerns and if you have further suggestions.
>
> [1] Tom Everitt and Marcus Hutter. Avoiding wireheading with value reinforcement learning. In Artificial General Intelligence, 2016.
> [2] Jacob Eisenstein, Chirag Nagpal, Alekh Agarwal, Ahmad Beirami, Alexander Nicholas D’Amour, Krishnamurthy Dj Dvijotham, Adam Fisch, Katherine A Heller, Stephen Robert Pfohl, Deepak Ramachandran, Peter Shaw, and Jonathan Berant. Helping or herding? reward model ensembles mitigate but do not eliminate reward hacking. In First Conference on Language Modeling, 2024.
> [3] Cassidy Laidlaw, Shivam Singhal, and Anca Dragan. Preventing reward hacking with occupancy measure regularization. In ICML Workshop on New Frontiers in Learning, Control, and Dynamical Systems, 2023.

---

> > ### Comment · Reviewer_bLnS · 2024-11-25
> >
> > Thank you for the authors' replies. I will keep my rating.

---

> > > ### Author Response · Authors · 2024-11-28
> > > **Official Reply to Reviewer bLnS**
> > >
> > > Thank you for taking the time to review our responses and for your thoughtful consideration of our work.
> > >
> > > Given your response, we were wondering what other modifications of our work you would like to see to increase the score. We would greatly appreciate any additional feedback on remaining weaknesses or concerns that you feel still need to be addressed.
> > >
> > > Thank you again for your time and efforts in reviewing our submission!

---

### Official Review · Reviewer_rZJE · 2024-11-05

**Soundness:** 1
**Presentation:** 1
**Contribution:** 2
**Rating:** 1
**Confidence:** 3

**Summary:**

This paper focuses on preventing reward hacking, in which optimizing a misspecified reward function causes undesired behavior. In particular, the authors focus on reward tampering, in which an agent can modify its reward function to receive higher reward with undesirable side effects. The authors propose a modification to DQN-like algorithms which decides whether to add newly collected transitions to the replay buffer depending on whether it seems like they will improve the current policy. They test the method, MC-DDQN, on some toy environments and find that it seems to prevent agents from reward hacking.

**Strengths:**

The paper focuses on an important problem and takes what appears to be a novel approach.

**Weaknesses:**

* **Mathematical correctness:** while the MC-DDQN algorithm is claimed to be grounded in General Utility RL, there are a number of issues with the presented algorithm that make it difficult to understand what it is actually doing. Consider Algorithm 2, for example. The ⇝ notation is already non-standard (e.g., it does not seem to be any of the uses [here](https://math.stackexchange.com/questions/680786/the-meaning-of-rightsquigarrow-in-math)). On line 2, the algorithm seems to suggest setting $U_{t+1}$ to $U^{\pi\_t}\_{VL\_t}$; however, on the first iteration of the for loop, $\pi\_t$ is not yet defined. On line 3, the algorithm refers to $\tau^\pi\_\rho$, which is not defined anywhere. In line 4, $\pi\_t$ takes as input the previous *transition* $T\_{t-1}$, despite policies being defined as taking states as input on line 133. The first part of line 5 should be $D \gets D \cup \\{ T\_{t - 1} \\}$. On the second part of line 5, $U\_{RL}$ has not been defined, only $F_{RL}$. Thus, *the core algorithm is completely unclear from the given description, given that most of the notation is undefined*.

 * **Limited experiments and evidence of usefulness:** given that the mathematical basis for the MC-DDQN algorithm is unclear, the remaining contribution of the paper lies in the experimental results. However, the experiments only focus on four gridworlds, and furthermore, in most of the experiments the proposed algorithm simply prevents the policy performance from dropping, which could just as easily be accomplished by freezing the policy when switching to the full environment.

Out of these two weaknesses, the first—that the main algorithm is ill defined—is the primary reason I rate the paper with a low score.

**Questions:**

How can Section 4 be reworked such that the concepts and algorithms presented are well-defined?

---

> ### Author Response · Authors · 2024-11-17
> **Official Reply to Reviewer rZJE [1/2]**
>
> Thank you for your review and valuable suggestions. In response to your points, we have provided detailed replies and made corresponding adjustments to our paper, including revising the notation and adding clarifications to Section 4. We also designed a new experiment to improve the empirical evaluation.
>
> > Several issues make it difficult to understand what the algorithm is actually doing
>
> Due to space constraints, we presented only a general algorithm description in the main text. To give readers a clearer understanding of what the algorithm is *actually* doing, we included a detailed description of our implementation in Appendix A.
>
> > The $\rightsquigarrow$ notation is non-standard
>
> The $\rightsquigarrow$ notation is used in Sutton & Barto [1] to describe Generalized Policy Iteration (GPI) in Chapter 4.6, page 86. We apply this notation to describe General Utility Generalized Policy Iteration (GU-GPI). The symbol conveys the notion of being driven toward, with the exact meaning depending on the algorithm implementing GPI, which we hope is clear from the main text.
>
> [1] Richard S Sutton and Andrew G Barto. Reinforcement learning: An introduction. MIT press, 2018.
>
> > On the first iteration of the for loop, $\pi_t$  is not yet defined
>
> The algorithm accepts an initial policy, $\pi_0$, as input, as stated on L218. We assume that at the first iteration of the for loop, $t=0$, so $\pi_t$ is defined and equal to $\pi_0$. We amended the paper to explicitly specify that $t$ starts with 0 to make the algorithm more clear.
>
> >  the algorithm refers to $\tau^\pi_\rho$, which is not defined anywhere
>
> The $\mathcal{T}^\pi_\rho$ is defined on line 201 and represents a distribution of trajectories beginning with state distribution $\rho$ and following policy $\pi$. We used $\tau_\rho^\pi$ as a random variable that follows this distribution. We adjusted the notation in Algorithm 1 and Algorithm 2 to clarify this.
>
> > $\pi_t$ takes as input the previous transition $T_{t−1}$, despite policies being defined as taking states as input
>
> On line 248, we explain that the state space is augmented to include the entire transition. We clarified this by changing the statement “The state space is augmented to include the full transition.” to  “The policy is adjusted to take the full transition as input, rather than just the environment state.”
>
> > The first part of line 5 should be $D \gets D \cup \\{ T_{t-1} \\}$
>
> Thank you for noticing this! We corrected this issue in the paper.
>
> > $U_\mathit{RL}$ has not been defined
>
> $U^\pi_\mathit{RL}$ is defined on line 197. In the algorithm, we initially omitted the superscript to avoid clutter, but we’ve reintroduced it to prevent any possible confusion.
>
> > The core algorithm is completely unclear from the given description, given that most of the notation is undefined.
>
> We hope that we clarified the unclear parts in our response. We also hope that our modifications to the paper will resolve these ambiguities for future readers.

---

> ### Author Response · Authors · 2024-11-17
> **Official Reply to Reviewer rZJE [2/2]**
>
> > Given that the mathematical basis for the MC-DDQN algorithm is unclear, the remaining contribution of the paper lies in the experimental results
>
> We hope that the clarifications above resolve some of your concerns and we would love to hear any additional feedback. We also see our contributions as going beyond the algorithm itself. Our work formalizes an agent with a learned utility function in the General Utility RL framework, connects reward hacking to inconsistent updates of the utility function, and proposes an evaluation protocol where the initial utility is learned in a safe environment from rewards.
>
> > The experiments only focus on four gridworlds
>
> Our experiments demonstrate that gridworlds, which prior literature discusses as examples of reward hacking, can be solved by avoiding inconsistent utility function updates. We emphasize that even these simple gridworld environments are unsolvable by previous approaches without constraining the policy from visiting new states. We are also adding an additional experiment, described in our common response. Please let us know whether it addresses this concern and whether you have any further suggestions.
>
> > In most of the experiments the proposed algorithm simply prevents the policy performance from dropping
>
> Our results show the algorithm’s ability to improve policy performance in three cases: performance increases in the Box Moving environment (Figure 3a), the Tomato Watering environment (Figure 3c), and the Absent Supervisor environment without walls (Figure 5d). Additionally, Figure 4b shows that our algorithm can be applied before the optimal policy is found in the safe version of the environment, with continued improvement in policy quality. In this case the policy also continues to improve.
>
> Thank you for highlighting your concerns about the clarity of our main algorithm. As you mentioned, this was your primary concern. We hope our clarifications address these points and provide a clear basis for evaluating the primary contributions of our work, including the algorithm’s correctness and novelty.

---

> ### Comment · Reviewer_rZJE · 2024-11-25
> **Response to rebuttal**
>
> Thank you to the authors for revising the notation and changing the algorithms to be more clear. I still find a lot of the method section quite unclear. While I understand the actual algorithm MC-DDQN implemented in practice, I think the mathematical notation is imprecise and potentially misleading. Here is a new (non-exhaustive) list of issues:
>
>  * How is $\mathcal{T}(D)$ precisely defined? It's described as "trajectories formed from the set of previously observed transitions $D$", but it would be good to have a formal definition. I assume it means something like
>
>    $$\mathcal{T}(D) = \\{ (s_0, a_0, \dots, s_h, a_h) \mid \forall t \in \\{0, \dots, h - 1\\} \quad \exists (s, a, s', r) \in D \quad \text{s.t.} \quad (s_h, a_h, s_{h + 1}) = (s, a, s') \\}$$
>
>  * On line 3 of both algorithms, do you mean $\\arg \\max_\\pi \\mathbb{E}\_{\\tau \\sim \\mathcal{T}\_\\rho^\\pi} [ U_{t + 1}(\\tau) ]$?
>
>  * On line 3 of both algorithms, how do you calculate $U_{t+1}(\tau)$ for a trajectory $\tau$ which is not in $\mathcal{T}(D)$? According to the rest of the algorithm, $U_{t+1}(\tau) = U^{\pi_t}_{VL_t} (\tau)$, which is only defined on Line 5 for $\tau \in \mathcal{T}(D)$.
>
> I'm also curious what the need is for considering GU-RL in the first place. Nowhere is there defined a fixed GU-RL utility function that MC-VL is optimizing.

---

> > ### Author Response · Authors · 2024-11-28
> > **Official Response to Reviewer rZJE**
> >
> > Thank you for your thoughtful consideration of our responses and for your continued engagement with our paper. We appreciate your feedback and are happy to incorporate it in the paper.
> >
> > > How is $\mathcal{T}(D)$ defined?
> >
> > Thank you for raising this question. The intuition behind $\mathcal{T}(D)$ is that current utility $U_{VL_t}$ is learned from the rewards in the previously observed data.  Since $U_{VL_t}$ is defined over a trajectory, we state that these trajectories are constructed from transitions previously observed and stored in $D$.
> >
> > We agree that including a formal definition can enhance clarity. Your proposed definition closely aligns with our intended meaning. We included the following definition in the main text: $$
> > \\mathcal{T}(D) = \\{ (s_0, a_0, \\dots, s_h, a_h)~\\forall t \\in \\{0, \\dots, h - 1\\}\\quad\\exists (s, a, s', r)\\quad\\in\\text{s.t.}\\quad(s_t, a_t, s_{t + 1}) = (s, a, s') \\}.
> > $$
> >
> > > Do you mean $\\arg \\max_\\pi \\mathbb{E}\_{\\tau \\sim \\mathcal{T}\_\\rho^\\pi} [ U\_{t + 1}(\\tau) ]$?
> >
> > Yes, thank you for the suggestion. We have updated the text in the paper to align with this notation.
> >
> > > How do you calculate $U\_{t+1}(\\tau)$ for a trajectory $\\tau$ which is not in $\\mathcal{T}(D)$?
> >
> > Thank you for this insightful question. The trajectory-value function $U\_t(\\tau)$ generalizes state-action value functions to trajectories and utilities. It can be evaluated on unseen trajectories, much like state-action value functions are applied to previously unseen states and actions.
> >
> > In Line 5, we describe how $U^{\\pi\_t}\_{VL\_t}$ is updated towards $U\_{RL}$ using previously observed data. The behavior of $U_t(\tau)$ on out-of-distribution trajectories depends on the specific implementation. In our case, $U_t(\tau)$ is parameterized as the sum of the reward model’s predictions for each transition and the state-action value function for the final transition. Both the reward model and the state-action value function are represented by neural networks. For states or actions not observed in the data, generalization relies on the interpolation and extrapolation capabilities of these networks.
> >
> > Additionally, our algorithm operates in an online setting, continuously collecting new data as the agent interacts with the environment. This iterative data collection helps correct potential errors in the trajectory-value function over time.
> >
> > > What is the need for considering GU-RL in the first place? Nowhere is there defined a fixed GU-RL utility function that MC-VL is optimizing
> >
> > MC-VL does indeed not optimize a single fixed utility function over the course of training. However, the policy optimizes a fixed utility function at each step of the algorithm. The use of GU-RL formalism was motivated by the need for a rigorous framework to describe utility functions and to compare policies (current and predicted) using these utilities. GU-RL provides precise definitions of utilities and policy optimization in an RL context, making it a natural choice. To facilitate policy comparison, we introduced trajectory-value functions, which assess the value of trajectories generated by different policies.
> >
> >  Our motivation for this work was to design an algorithm robust enough to perform well even under an "optimal oracle" policy—one capable of accurately predicting the effects of actions on both the environment and the agent’s learning process. For example, in standard RL with a fixed utility function $U_{\mathit{RL}}$, an optimal policy might tamper with rewards to maximize the reward signal throughout the decision horizon. To mitigate this, our algorithm optimizes the current utility at each timestep. Since for a lot of tasks, utility is hard to correctly specify, we learn it from the observed rewards, treating them as evidence for true utility.
> >
> > This approach discourages reward tampering because the learned utility is unlikely to assign high value to trajectories with reward tampering. Past experience typically rewards actions aligned with the task’s intended objectives, and learning from reward tampering trajectories would likely result in changes in the behavior that are evaluated poorly by the current utility, discouraging the oracle policy from actions that lead to such outcomes. We believe it is essential to develop algorithms that remain robust under such oracle policies as we move toward creating more general and capable agents.
> >
> > In summary, while our algorithm does not optimize a single utility function throughout training, it optimizes a fixed utility at each step, ensuring consistent updates that mitigate reward hacking in the environments we studied. The GU-RL formalism provided the language necessary to describe these utilities and their evolution during training, enabling us to develop a framework that addresses reward hacking in the environments presented in our experiments.
> >
> > Thank you again for your thorough and valuable review!

---

### Official Review · Reviewer_VGrY · 2024-11-06

**Soundness:** 3
**Presentation:** 2
**Contribution:** 2
**Rating:** 5
**Confidence:** 3

**Summary:**

This paper explores how modifying the utility function’s design could mitigate the issue of reward hacking in RL. It introduces a new algorithm called MC-VL, aimed at addressing cases where RL agents maximize reward signals by exploiting incomplete or ambiguous reward functions, potentially misaligning with the designer's actual goals.

**Strengths:**

1. This paper proposes a novel approach to address inconsistencies during the learning process.
2. It discusses reward hacking within the framework of GU-RL, providing in-depth insights and theoretical support for tackling this issue.

**Weaknesses:**

- The paper notes that MC-VL requires predictions about future behaviors, which could entail higher computational costs and may limit its applicability in more complex environments.
- The proposed method assumes that the agent can learn the correct utility function from observed rewards. In more complex or uncertain environments, the performance of the algorithm could suffer if the reward model fails to generalize accurately to new trajectories.
- The experiments are primarily conducted in relatively simple grid-world environments, leaving the algorithm’s effectiveness in more complex, high-dimensional environments still unverified.

**Questions:**

1. Were all experiments conducted by training on the "Safe version" before transitioning to the "Full version"? If so, how would the proposed algorithm generalize to typical environments?
2. The authors mention that hyperparameters significantly influence algorithm performance, listing many in Table 1 of Appendix E, such as "Inconsistency check training steps l," "Inconsistency check rollout steps h," "Number of inconsistency check rollouts k," and "Predicted reward difference threshold." However, it’s unclear how these parameters specifically impact the proposed algorithm or how they should be selected.
3. Reward hacking is common across various environments, as mentioned in the robotic context at the paper's beginning. Could this algorithm potentially be applied to continuous action versions, like TD3 for DDQN? If extended to continuous environments, would the accuracy of the utility function become more critical to the algorithm's performance?
4. Could the authors validate the algorithm's effectiveness in more complex environments, such as those with high-dimensional inputs (e.g., continuous or image-based inputs)?
5. check the definnation of $\lambda(\tau)$

---

> ### Author Response · Authors · 2024-11-16
> **Official Reply to Reviewer VGrY [1/3]**
>
> Thank you very much for your thorough review and valuable suggestions. In response to each of your points, we are providing detailed replies below. We also made corresponding adjustments to our paper including revising the Limitations section and Appendix E. Moreover, we designed a new experiment to improve the empirical evaluation.
>
> >  MC-VL requires predictions about future behaviors, which could entail higher computational costs
>
> We agree with the reviewer that these additional predictions about future behavior come at an additional computational cost. Reducing this computational cost has not been our focus in this paper and is an interesting area of research. Instead, we want to establish the feasibility of precluding reward hacking during policy learning. We hypothesize that future advances in RL (such as through in-context RL) will allow to minimize the overhead of future behaviour prediction without costly additional training [1]. We revised the Limitations section to reflect this expectation more accurately.
>
> [1] Jakob Bauer, Kate Baumli, Feryal Behbahani, Avishkar Bhoopchand, Nathalie Bradley-Schmieg, Michael Chang, Natalie Clay, Adrian Collister, Vibhavari Dasagi, Lucy Gonzalez, Karol Gregor, Edward Hughes, Sheleem Kashem, Maria Loks-Thompson, Hannah Openshaw, Jack Parker-Holder, Shreya Pathak, Nicolas Perez-Nieves, Nemanja Rakicevic, Tim Rocktäschel, Yannick Schroecker, Satinder Singh, Jakub Sygnowski, Karl Tuyls, Sarah York, Alexander Zacherl, and Lei M Zhang.
> Human-timescale adaptation in an open-ended task space. In ICML, 2023.
>
>
> > The performance of the algorithm could suffer if the reward model fails to generalize accurately to new trajectories
>
> We agree, our approach could suffer if the reward model completely fails to generalize. However, we do not require the reward model to be perfectly accurate, it only needs to be accurate enough to distinguish between normal and reward hacking behaviors. Moreover, learning an accurate reward model is generally simpler than learning a precise value function or policy. It is a supervised task with a fixed target throughout training. Our method leverages the reward model to validate the consistency of an updated policy, rejecting updates that lead to previously undesirable behaviors. We believe this approach should improve the resulting system safety.
>
> > Were all experiments conducted by training on the "Safe version" before transitioning to the "Full version"? If so, how would the proposed algorithm generalize to typical environments?
>
> Yes, we indeed trained on the *Safe* version first. We consider safety crucial for real-world interactions and discuss several ways to set up *Safe* environments in our paragraph on Initial Utility Function (L276-285). For example, simulation training can serve as a *Safe* version before sim-to-real transfer. This is a common workflow in RL already [2]. Alternatively, a *Safe* controlled lab environment can allow monitoring for reward hacking and resetting the experiment as needed, which we also believe to be the case for many typical deployment processes.
>
> [2] W. Zhao, J. P. Queralta and T. Westerlund, "Sim-to-Real Transfer in Deep Reinforcement Learning for Robotics: a Survey," 2020 IEEE Symposium Series on Computational Intelligence (SSCI), Canberra, ACT, Australia, 2020, pp. 737-744, doi: 10.1109/SSCI47803.2020.9308468.

---

> ### Author Response · Authors · 2024-11-17
> **Official Reply to Reviewer VGrY [2/3]**
>
> >  It’s unclear how certain hyperparameters impact the proposed algorithm or how they should be selected
>
> Thank you for highlighting this concern. Unfortunately, our computational budget does not allow for an exhaustive hyperparameter search. We provide some guidance for choosing hyperparameters in Appendix C, and specific adjustments for environments in Appendix E.1. We also analyze the impact of two key hyperparameters:
> - The effect of **changing inconsistency check training steps $l$** is shown in Figure 5a. Our findings suggest that if $l$ is too low, the method becomes less effective in preventing reward hacking, with gradual performance decay. For values larger than 500, preliminary experiments showed consistent performance.
> - The effect of **removing the Predicted reward difference threshold** is shown in Figure 4a. As discussed in the Ablations and Sensitivity Analysis section (L463-466), removing this hyperparameter does not impact performance, with its primary role being to improve computational efficiency.
>
> We have expanded the hyperparameter descriptions and guidelines in Appendix E to enhance clarity:
> - **Inconsistency check training steps $l$** $~~~~$   This parameter describes the number of updates to the Q-function needed to predict the future policy based on a new transition. As shown in Figure 5a, this value must be sufficiently large to update the learned values and corresponding policy. It can be selected by artificially adding a transition that alters the optimal policy and observing the number of training steps required to learn the new policy.
> - **Inconsistency check rollout steps $h$**  $~~~~$  This parameter controls the length of the trajectories used to compare two predicted policies. The trajectory length must be adequate to reveal behavioral differences between the policies. In this paper, we used a fixed, sufficiently large number. In episodic tasks, a safe choice is the maximum episode length; in continuing tasks, a truncation horizon typically used in training may be suitable. Computational costs can be reduced by choosing a smaller value based on domain knowledge.
> - **Number of inconsistency check rollouts $k$** $~~~~$ This parameter specifies the number of trajectories obtained by rolling out each predicted policy for comparison. The required number depends on the stochasticity of the environment and policies. If both the policy and environment are deterministic, $k$ can be set to 1. Otherwise, $k$ can be selected using domain knowledge or replaced by employing a statistical significance test.
> - **Predicted reward difference threshold**  $~~~~$  This threshold defines the minimum difference between the predicted and observed rewards for a transition to trigger an inconsistency check. As discussed in Section 5.4, this parameter does not impact the algorithm’s performance and can be set to 0. However, it can be adjusted based on domain knowledge to speed up training by minimizing unnecessary checks. The key requirement is that any reward hacking behavior must increase the reward by more than this threshold relative to the reward predicted by the reward model.
>
> > Could this algorithm potentially be applied to continuous action versions, like TD3 for DDQN?
>
> Yes, it can be applied to continuous action spaces. As noted in the common response, we aim to validate the algorithm's applicability to continuous action spaces using TD3 in our additional experiment.
>
> > If extended to continuous environments, would the accuracy of the utility function become more critical to the algorithm's performance?
>
> In our experiments, the utility function is parametrized as the sum of rewards along a trajectory and the Q-value of the final state. With sufficiently high rollout steps $h$, the utility function's accuracy largely depends on the reward function's accuracy, which we do not expect to be significantly impacted in continuous action environments. However, estimating future behavior from a single transition can become more challenging with continuous action spaces. We expect it to encounter the same issues as offline RL, and mitigating these issues remains an active area of research.

---

> ### Author Response · Authors · 2024-11-17
> **Official Reply to Reviewer VGrY [3/3]**
>
> > Could the authors validate the algorithm's effectiveness in more complex environments, such as those with high-dimensional inputs (e.g., continuous or image-based inputs)?
>
> This is a shared concern among reviewers. To address this, we propose an additional experiment in our common response, where we will test the algorithm in the environment with continuous inputs and actions. We welcome your feedback on our proposed approach and additional experiment to address this concern.
>
> > Check the definition of $\lambda(\tau)$.
>
> Our definition of $\lambda(\tau)$ closely follows the one provided by [3] in Equation 8. We made the adjustments to accommodate our notation, which uses a different indexing scheme to simplify expressions.
>
> [3] Anas Barakat, Ilyas Fatkhullin, and Niao He. Reinforcement learning with general utilities: Simpler variance reduction and large state-action space. In ICML, 2023
>
> Thank you again for your detailed feedback! Please let us know if you find our responses satisfactory and if you have any further suggestions.

---

### Author Response · Authors · 2024-11-16
**Common Response**

We thank the reviewers for their thoughtful feedback! We are encouraged that reviewers find our work novel (**VGrY**, **rZJE**, **bLnS**) and recognize that it tackles an important problem  (**rZJE**, **bLnS**, **oLm1**). We are grateful for your praise of it as interesting (**bLnS**), insightful (**VGrY**), and elegant (**oLm1**). As we are working on improving the paper and running additional experiments, we would like to use this initial response to addressing the reviewer’s points and ask for some clarifications or additional feedback. The responses provided here and the additional experiments will also be incorporated in the paper. We understand the reviewers' concern about the fact that our algorithm is evaluated only in grid-world environments. Reviewers also asked if our algorithm can be applied to continuous action spaces (**VGrY**, **oLm1**) and to implement our MC-VL algorithm based on other RL methods, such as TD3 or DDPG (**VGrY**, **bLnS**).

We decided to make our best effort to address the reviewers’ concerns within our computational constraints. We are currently implementing a version of our algorithm based on **TD3**, which will be applicable to continuous action spaces. We are planning to evaluate it on the environment based on *MuJoCo Reacher-v5*. The environment contains a robotic arm, and the goal is to reach a randomly selected goal position. The reward is a sum of negative distance to the goal and negative torques applied to the joints. To study reward hacking, we modify the environment by introducing an invisible button which provides a maximum possible reward if the arm holds it for several consecutive frames. We hope to address several concerns with this experiment:
- Show that MC-VL can be implemented based on a different RL method (TD3).
- Verify that our algorithm can work with continuous action spaces.
- Provide evaluation in a more complex environment.

We want to note that our work, to the best of our knowledge, is first to show empirical evidence for the possibility of mitigation of reward hacking without constraining the policy to be close to a known safe policy or requiring a prior over manually specified utility functions, even in the grid-world setting. Our primary objective is not providing a practical and ready-to-use solution to reward hacking in all environments, but showing that such a solution is possible in cases when reward hacking is caused by an inconsistent update to the utility function. Our experiments were designed to illustrate that such inconsistency may arise in several environments discussed in the literature, and avoiding it prevents the agent from reward hacking.

However, we agree that more extensive empirical evaluation is always a good thing. We would love to hear the reviewer’s feedback on this additional experiment and hope that it will address any concerns along with the clarifications in the replies below.

---

### Author Response · Authors · 2024-11-22
**General Response #2 [1/2]**

We are excited to share that we were able to complete the additional experiment requested in the review process. The new agent based on TD3, which we call MC-TD3, successfully trained in the modified MuJoCo Reacher-v5 environment while avoiding reward hacking. This environment involves a reaching task with a robotic arm following the suggestions by reviewers bLnS and oLm1. We sincerely thank the reviewers for their valuable feedback, which motivated this experiment and has significantly strengthened our empirical evaluation. This experiment demonstrates that the MC-VL algorithm is both general enough to be integrated with other RL algorithms and robust enough to perform effectively in complex environments with continuous action and observation spaces. Additionally, it shows that our approach works even when the initial utility function is learned from transitions collected during random exploration, as long as those transitions do not include reward hacking behaviors.

To reflect this new contribution, we have updated the paper. To simplify the reviewers' work, we provide all information on the new experiment below, and we also attached a LaTeXdiff version of the paper showing the differences to the initially submitted draft at the end of the updated manuscript. The material related to the newly added experiment contains the following key additions and revisions:

- In Section 5.1 (Environments) we added a paragraph describing the Reacher environment:

> **Reacher** This environment is based on the Reacher-v5 environment from the MuJoCo subset of the Gymnasium benchmark (Towers et al., 2024). It involves a robotic arm tasked with reaching a randomly placed target, starting from random joint rotations and velocities. We modified the environment by adding an invisible button that provides a one-time reward of +50 when held for 15 consecutive steps, simulating reward tampering. The performance metric does not include this reward. Further details are provided in Appendix G.
- We added Figure 3.e. which demonstrates the performance of MC-TD3 in the Reacher environment. The plot shows that the performance of MC-TD3 steadily improves throughout the training. The returns of the TD3 baseline grow, but the performance metric drops, indicating reward hacking.
- We added Appendix E.3, which contains the hyperparameters of MC-TD3.

---

> ### Author Response · Authors · 2024-11-22
> **General Response #2 [2/2]**
>
> - We added Appendix F which describes the implementation of MC-TD3 as follows:
>
>   > Our implementation is based on the implementation provided by Huang et al. (2022). The overall structure of the algorithm is consistent with MC-DDQN, described in  Appendix A, with key differences outlined below. TD3 is an actor-critic algorithm, meaning that the parameters $\theta$ define both a policy (actor) and a Q-function (critic). In  Algorithm 3 and Algorithm 5, calls to *TrainDDQN* are replaced with *TrainTD3*, which updates the actor and critic parameters $\theta$ as specified by Fujimoto et al. (2018). Furthermore, in Algorithm 3, the returned policy $\pi_f(s)$ corresponds to the actor rather than $\arg\max_a \dot{Q}(s, a; \theta_f)$ and in Algorithm 5 the action executed in the environment is also selected by the actor.
>
> - We added Appendix G describing the details of the experiment in the Reacher environment:
> > The rewards in the original Reacher-v5 environment are calculated as the sum of the negative distance to the target and the negative joint actuation strength. This reward structure encourages the robotic arm to reach the target while minimizing large, energy-intensive actions. The target's position is randomized at the start of each episode, and random noise is added to the joint rotations and velocities. Observations include the angles and angular velocities of each joint, the target's coordinates, and the difference between the target's coordinates and the coordinates of the arm's end. Actions consist of torques applied to the joints, and each episode is truncated after 50 steps.
> We modified the environment by introducing a +50 reward when the arm's end remains within a small, fixed region for 15 consecutive steps. This region remains unchanged across episodes, simulating a scenario where the robot can tamper with its reward function, but such behavior is difficult to discover. In our setup, a reward-tampering policy is highly unlikely to emerge through random actions and is typically discovered only when the target happens to be near the reward-tampering region.
> In accordance with standard practice, each training run begins with exploration using random policy. For this experiment, we do not need a separate *Safe* environment; instead, the initial utility function is trained using transitions collected during random exploration. This demonstrates that our algorithm can function effectively even when a *Safe* environment is unavailable, provided that the initial utility function is learned from transitions that do not include reward hacking.
>
> We hope these updates address the reviewers’ concerns and illustrate the robustness and generality of our approach. Please let us know if you have any more questions or require any further changes. If this alleviates some of your raised concerns, please  consider raising the scores to support acceptance of the paper. Thank you for your thoughtful feedback and the opportunity to improve our work.

---

### Meta-Review · Area_Chair_2kRa · 2024-12-20

**Metareview:**

This paper aims to create a method to reduce the prevalence of reward hacking, which makes designing easy-to-optimize rewards significantly challenging. The paper's proposed method is to leverage a utility function (a reward for a whole trajectory) and a corresponding value function, then augment the action space with an additional action to modify the utility function. This modification allows the agent to learn if changing the utility functions leads to better learning. The hope of this method is that agent can learn when learning from certain transitions lead to reward hacking.

As noted by the reviewers, the paper lacks mathematical precision, which makes the method unclear and prevents a sufficient understanding of the paper's contribution. The reviewers also note concern about the limit of using four gridworld environments in the experiments. I do not share this same concern by default, but the paper's motivation is to alleviate reward hacking in more complex environments. As such, the paper should make it clear what is learned from these experiments in pursuit of this direction.

**Additional Comments On Reviewer Discussion:**

The reviewers and authors discussed issues related to lack of clarity and mathematical precision. Some issues were resolved, but many remained. Based on this discussion, the paper warrants further revision to address this issue before publication.

---

### Decision · Program_Chairs · 2025-01-22

Reject